# ERAD-dependent control of the Wnt secretory factor Evi

Kathrin Glaeser[1], Manuela Urban[1], Emma Fenech[2], Oksana Voloshanenko[1], Dominique Kranz[1], Federica Lari[2], John C Christianson[2] (iD) & Michael Boutros[1,*] (iD)

## Abstract

**Active regulation of protein abundance is an essential strategy to modulate cellular signaling pathways. Within the Wnt signaling cascade, regulated degradation of β-catenin by the ubiquitin-proteasome system (UPS) affects the outcome of canonical Wnt signaling. Here, we found that abundance of the Wnt cargo receptor Evi (Wls/GPR177), which is required for Wnt protein secretion, is also regulated by the UPS through endoplasmic reticulum (ER)-associated degradation (ERAD). In the absence of Wnt ligands, Evi is ubiquitinated and targeted for ERAD in a VCP-dependent manner. Ubiquitination of Evi involves the E2-conjugating enzyme UBE2J2 and the E3-ligase CGRRF1. Furthermore, we show that a triaging complex of Porcn and VCP determines whether Evi enters the secretory or the ERAD pathway. In this way, ERAD-dependent control of Evi availability impacts the scale of Wnt protein secretion by adjusting the amount of Evi to meet the requirement of Wnt protein export. As Wnt and Evi protein levels are often dysregulated in cancer, targeting regulatory ERAD components might be a useful approach for therapeutic interventions.**

**Keywords** CGRRF1; Evi/Wls/GPR177; Porcn; regulatory ERAD; Wnt signaling
**Subject Categories** Membrane & Intracellular Transport; Protein Biosynthesis & Quality Control; Signal Transduction
**The EMBO Journal (2018) 37: e97311**

## Introduction

Post-translational protein regulation is an important mechanism that provides cells with rapid responses to changes in intra-and extra-cellular conditions (Lee & Yaffe, 2016). Steady-state protein levels are controlled by changes in transcription and translation as well as by their intrinsic stability, which is affected by folding state, localization, and co-assembly with partner proteins. For proteins of the secretory pathway, endoplasmic reticulum (ER)-associated degradation (ERAD) is a key cellular process to ensure protein quality and quantity control (Hegde & Ploegh, 2010). ERAD is a specialized ubiquitin-proteasome system (UPS)-dependent process that prevents the secretion and aggregation of proteins that failed to fold or assemble appropriately (Vembar & Brodsky, 2008). Following coordinated recognition and delivery to membrane-embedded ubiquitination machineries, ERAD substrates are retrotranslocated across the ER membrane into the cytosol through the AAA-ATPase VCP (also known as p97) and degraded by 26S proteasomes (Ye et al, 2001; Vembar & Brodsky, 2008; Smith et al, 2011; Olzmann et al, 2015). Since ERAD-dependent quality control has been linked to a range of cellular processes and human diseases (Zettl et al, 2011; Guerriero & Brodsky, 2012; Perrody et al, 2016), understanding its molecular mechanisms is an important step to develop potential treatment strategies (Tsai & Weissman, 2010; Hetz et al, 2013). In addition to its role in protein quality control of nascent polypeptides, ERAD has been implicated in regulating the abundance of mature proteins in response to changes in physiological conditions (Wiertz et al, 1996a,b; Sever et al, 2003; Brodsky & Fisher, 2008; Adle et al, 2009; Foresti et al, 2013; Avci et al, 2014; van den Boomen et al, 2014). However, only a few regulatory ERAD substrates have been described so far. Given the number of ubiquitin ligases present in the ER (Neutzner et al, 2011), the physiological role of regulatory ERAD is likely underestimated at present.

Wnt signaling is crucial during development and adult tissue homeostasis (Logan & Nusse, 2004), and its aberrant regulation has been linked to many diseases, including cancer (Clevers & Nusse, 2012; Zhan et al, 2017). Nineteen Wnt ligands are produced and subsequently processed in the endoplasmic reticulum (ER) by the attachment of palmitoleic acid to a conserved serine residue by the acyltransferase Porcupine (Porcn) (Kadowaki et al, 1996; Willert et al, 2003; Takada et al, 2006). In the Wnt secretory pathway, this lipid modification is important for the interaction between Wnt proteins and the conserved transmembrane protein Evi (Wls, GPR177), which shuttles lipid-modified Wnt proteins to the plasma membrane and onto exovesicles (Bänziger et al, 2006; Bartscherer et al, 2006; Goodman et al, 2006; Gross et al, 2012; Herr & Basler, 2012). Evi is then recycled via the retromer complex back to the Golgi and ER (Coudreuse, 2006; Belenkaya et al, 2008; Franch-Marro et al, 2008; Pan et al, 2008; Port et al, 2008; Yang et al, 2008; Yu et al, 2014). The

1  Division of Signaling and Functional Genomics, German Cancer Research Center (DKFZ) and Department of Cell and Molecular Biology, Medical Faculty Mannheim, Heidelberg University, Heidelberg, Germany
2  Ludwig Institute for Cancer Research, University of Oxford, Oxford, UK
   *Corresponding author. Tel: +49 6221 421950; E-mail: m.boutros@dkfz.de

levels of both Wnt and Evi proteins are often aberrantly regulated in cancer (Augustin *et al*, 2012; Voloshanenko *et al*, 2013; Stewart *et al*, 2015), which contributes to high Wnt activity during tumorigenesis.

Here, we identified a regulatory ERAD mechanism that continuously removes the Wnt shuttling receptor Evi from the ER in the absence of lipid-modified Wnt proteins. Poly-ubiquitination and degradation of Evi involve the E2 ubiquitin-conjugating enzyme UBE2J2, the ubiquitin ligase CGRRF1, and the AAA-ATPase VCP. Responding to Wnt protein availability in the ER, a triaging complex containing Porcn and VCP directs Evi either toward ERAD or the secretory pathway. Since expression of different Wnt proteins prevents ERAD-dependent degradation of Evi, our data demonstrate that Wnt-induced Evi stabilization automatically adjusts the amount of Evi to accommodate secretion of Wnt ligands. Control of Evi abundance through ERAD represents a cellular feedback mechanism to control Wnt protein secretion.

## Results

### Wnt proteins stabilize the cargo receptor Evi

Increased Evi protein abundance coincides with elevated Wnt3 expression in colorectal cancer (Fig EV1A; TCGA, 2012; Voloshanenko *et al*, 2013). However, transcriptome profiles of 456 colon adenocarcinoma samples showed no increase in Evi mRNA levels (Fig EV1B; TCGA, 2012). Similarly, immunohistochemistry and *in situ* RNA hybridization carried out on sequential tissue sections found cases of colon carcinoma where increased Evi protein was not matched by a concomitant increase in Evi mRNA (Fig 1A; Appendix Fig S1A and B). These observations raised the question how Evi protein levels might be regulated under physiological conditions.

Since several Wnt proteins are upregulated in colorectal cancer (TCGA, 2012; Voloshanenko *et al*, 2013), we assumed that an increase in Wnt proteins might affect the levels of the cargo receptor Evi, allowing the cells to adjust to the molecular requirements for Wnt secretion. To test this hypothesis, we transiently increased Wnt levels in HEK293T cells and observed an elevation in Evi protein levels upon Wnt3A expression, but not upon expression of a secreted control protein, IGFBP5-V5 (Fig 1B). The Wnt-regulated protein bands were specific for Evi since they were absent in Evi

knockout (Evi$^{KO}$) HEK293T cells (Fig 1B; Appendix Fig S2). Wnt-induced increase in protein abundance appeared to be selective for Evi since levels of the Wnt co-receptor LRP6 were not affected by Wnt3A expression (Fig 1B).

The human genome encodes 19 different Wnt proteins (van Amerongen & Nusse, 2009) and so we asked next whether also other Wnts are capable to increase Evi protein levels. Expression of different canonical and non-canonical Wnt ligands strongly increased Evi protein levels, which was not observed upon expression of secreted luciferase (sLuc) or IGFBP5 (Fig 1C). Overexpression of Wnt11 led to only a modest increase in Evi levels. To address whether Wnt ligands elevated Evi levels by inducing the intracellular Wnt signaling cascade, we monitored Evi not only upon Wnt3A expression, but also upon activation of the Wnt cascade below the receptor level by expressing the Wnt activator Dishevelled2 (Dvl2) or by blocking GSK3β activity with the small molecule SB216763 (Fig 1E). While the Wnt target gene AXIN2 was upregulated under all conditions (Fig EV1C), Evi protein levels increased only upon Wnt3A expression (Fig 1D). These experiments suggest that the increase in Evi protein levels is specifically mediated by Wnt ligands and not by Wnt pathway activation (Fig 1E). Notably, exogenously provided recombinant Wnt3A was not sufficient to increase Evi protein levels (Fig 1D, lanes 5–8), indicating that cell-autonomous Wnt protein production is required.

The Wnt-dependent increase in Evi protein abundance could be attributed to upregulated transcription of Evi. However, we did not observe changes in Evi mRNA levels upon Wnt3A expression (Fig EV1C), which is consistent with a lack of increased Evi expression in colorectal cancer (Figs 1A and EV1B). To confirm that the observed increase in Evi protein levels was not due to transcriptional changes upon canonical Wnt pathway activation, we monitored Evi levels following knockdown of β-catenin. Upon Wnt3A expression, β-catenin knockdown reduced canonical Wnt reporter activity to basal levels (Fig EV1D′) but did not affect the Wnt-induced increase in Evi protein (Fig EV1D). Taken together, these results support a model whereby Evi protein levels are post-transcriptionally regulated by the concomitant secretion of Wnt proteins (Fig 1E).

### Evi regulation depends on Wnt palmitoylation

Wnt proteins are palmitoylated at a conserved serine residue by the ER-resident O-acyltransferase Porcn (Kadowaki *et al*, 1996; Willert

---

**Figure 1.  Wnt ligand production increases Evi protein levels.**

A  *In situ* RNA hybridization (red dots) and immunohistochemistry (brown staining) of Evi were performed on sequential FFPE tissue slides of healthy colon and matched colon cancer tissue from five patients. The illustrated example is representative for three patients. Scale bar: 40 μm. Specificity of Evi probes was confirmed in Appendix Fig S1B.

B  Wild-type (wt) or Evi$^{KO}$ HEK293T cells were transfected with Wnt3A or IGFBP5-V5 expression plasmids and subjected to Western blot analysis. Specific Evi bands are indicated by arrows and unspecific bands by asterisks. Endogenous Evi is not only detectable as a monomeric form (46 kDa) but also as SDS-resistant dimers (80 kDa). Clonal Evi$^{KO}$ HEK293T cells were generated via CRISPR/Cas9 using Evi sgRNA #2 (Evi$^{KO2.9}$) or Evi sgRNA #1 (Evi$^{KO1.1}$; Appendix Fig S2).

C  HEK293T cells were transfected with Wnt expression plasmids and analyzed for endogenous Evi levels by immunoblotting with a mouse monoclonal Evi antibody (Biolegend, clone YJ5).

D  HEK293T cells were transfected with the indicated overexpression constructs, treated with 100 ng/ml recombinant mouse Wnt3A (rec. W3A) or with 10 μM GSK3β inhibitor SB216763 for the indicated hours (h). The obtained cell lysates were used for Western blot analysis. Representative Western blots of three independent experiments are shown. β-Actin or N-cadherin were used as a loading control, and LRP6 served as a reference membrane protein.

E  Scheme showing that Evi is regulated through Wnt proteins within the Wnt-producing cell. Canonical Wnt signaling can be activated by Wnt ligands, Dishevelled (Dvl) overexpression or by the GSK3β inhibitor SB216763.

Source data are available online for this figure.

*et al*, 2003; Takada *et al*, 2006), which is required for the Wnt proteins to interact with Evi (Herr & Basler, 2012) (Fig 2A). To determine whether Wnt palmitoylation and consequently Evi-Wnt

interactions are required for the Wnt-mediated increase in Evi protein, we treated wild-type, Wnt3, or Wnt5B stable transfected HEK293T cells with the Porcn inhibitor LGK974 to block Wnt

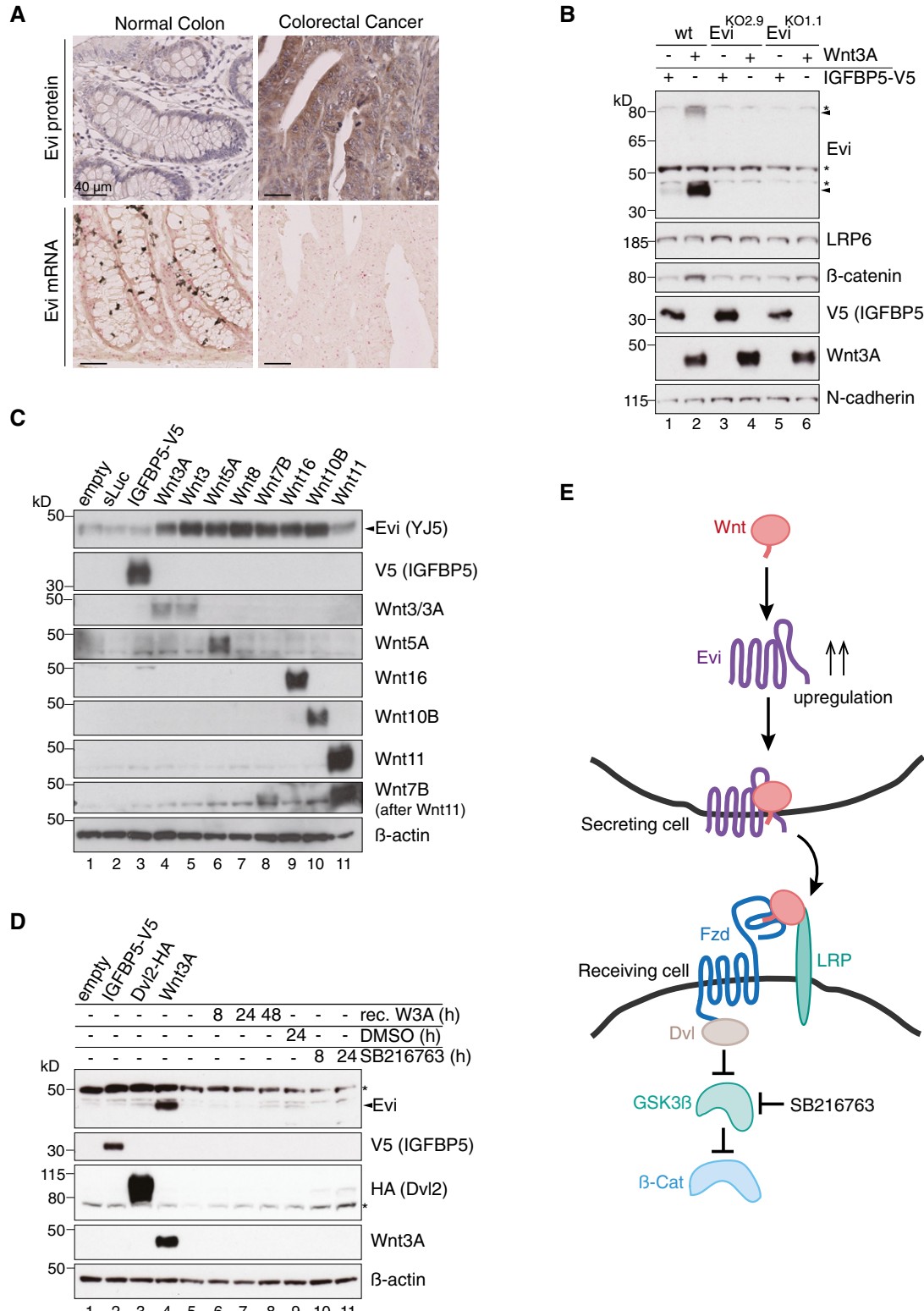

**Figure 1.**

palmitoylation. Strikingly, the Wnt-induced increase in Evi protein levels was blocked upon Porcn inhibition, indicating that Porcn activity is required for Wnt-mediated Evi regulation (Fig 2B). In contrast to the Wnt target gene AXIN2, mRNA levels of Evi were not affected by LGK974 treatment indicating a post-transcriptional regulation of Evi (Fig EV2A).

Porcn knockout (Porcn$^{KO}$) HEK293T cells were generated (Appendix Fig S3) to confirm the importance of Porcn for Wnt-mediated Evi regulation. Unlike wild-type HEK293T cells, Porcn$^{KO}$ cells were not responsive to Wnt3A expression despite retaining intact Wnt signaling capability as confirmed by Wnt pathway activation upon Dvl3 overexpression (Fig EV2B). Using these cell lines, we found that Wnt3A expression led to an increase in Evi protein levels in wild-type but not in Porcn$^{KO}$ HEK293T cells (Fig 2C).

To investigate whether the reduced steady-state levels of Evi upon Porcn inhibition are caused by the lack of Wnt palmitoylation, HEK293T cells were transfected with the palmitoylation mutant Wnt3A S209A and several wild-type Wnt3A constructs. All tested Wnt3A variants stabilized Evi with the exception of Wnt3A S209A, which produced a phenotype similar to that of Porcn inhibition (Fig 2D).

To confirm the role of Porcn function in Evi regulation without manipulating Wnt expression, we monitored Evi in the colon cancer cell line HCT116, that depends on Wnt secretion (Voloshanenko *et al*, 2013) and in the melanoma cell line A375, that expresses Wnt5A and Wnt10B endogenously (Yang *et al*, 2012). LGK974 treatment of HCT116 and A375 cells reduced Evi protein levels in a time-dependent manner and depleted Evi below the detection limit after 24–48 h (Fig 2E) without reducing Evi mRNA levels (Fig EV2C and D). These results demonstrate that the palmitoylation of Wnts by Porcn is required for the Wnt-induced accumulation of Evi (Fig 2F).

## Evi is poly-ubiquitinated and degraded by the proteasome

We next asked whether palmitoylated Wnt proteins increase Evi protein levels by altering its rate of degradation. For this purpose, Evi levels were monitored following impairment of the lysosomal and proteasomal degradation pathways. Blocking lysosomal degradation by treatment with the V-ATPase inhibitor Bafilomycin A increased Evi protein levels independent of Wnt3A expression (Fig EV3A). In contrast, inhibition of proteasomal degradation by treatment with the proteasome inhibitors MG132 (Fig 3A) or bortezomib (Fig EV3B) increased Evi protein levels in the absence of

Wnt3A, while no additional increase was observed when Wnt3A was expressed. The presence of cell-autonomous Wnt proteins thus seems to be sufficient to prevent proteasome-dependent degradation of Evi, promoting its stability.

Proteins targeted for proteasomal degradation are conventionally marked by covalent poly-ubiquitin chains. Using ubiquitin-affinity precipitation from lysates of MG132-treated cells, we were able to detect poly-ubiquitination of endogenous Evi (Fig 3B). Treatment with the deubiquitinating enzyme (DUB) Usp2 reduced the signal of poly-ubiquitinated Evi (Fig EV3C), confirming direct ubiquitin-modification of Evi. Notably, the amount of poly-ubiquitinated Evi was reduced upon Wnt3 expression (Fig 3B, lane 4), supporting the ability of Wnts to diminish the ubiquitination and thus proteasomal degradation of Evi (Fig 3C).

Proteasomal degradation of integral membrane proteins like Evi in the early secretory pathway could be mediated through ERAD (Vembar & Brodsky, 2008). To ascertain whether Wnts stabilize Evi within the ER, we fused a KDEL retrieval sequence to Wnt3A (Wnt3A-KDEL) to restrict its trafficking (Fig EV3D) and monitored changes in Evi abundance. We found that the expression of ER-retained Wnt3A-KDEL stabilized Evi protein levels, which was blocked by LGK974 treatment (Fig 3D). These results confirm that palmitoylated Wnt proteins stabilize Evi within the ER.

## The AAA-ATPase VCP is required for Evi degradation

Proteasome-dependent degradation from the ER in the absence of Wnt proteins suggests that Evi might be constitutively targeted for ERAD. To determine whether Evi is an endogenous ERAD substrate, we silenced the expression of several known ERAD components and monitored changes in Evi abundance. The AAA-ATPase VCP is an essential component of ERAD processing by promoting the dislocation of ERAD substrates across the ER membrane into the cytosol to allow proteasomal degradation (Ye *et al*, 2001; Fig 4A). Knockdown of VCP by both pooled and single siRNAs stabilized endogenous Evi protein without affecting its mRNA levels (Figs 4B and EV3E). Similarly, treating HEK293T cells with the specific allosteric VCP inhibitor NMS-873 (Magnaghi *et al*, 2013) stabilized Evi in a concentration-dependent manner (Fig 4C). Multiple Evi-immunoreactive bands at higher molecular weights in the absence of VCP activity are consistent with the accumulation of poly-ubiquitinated species and impaired delivery to the proteasome.

---

**Figure 2.  Evi stabilization is dependent on Wnt palmitoylation.**

A  Schematic illustration of the Porcn-mediated Wnt palmitoylation, which is important for Evi-Wnt interaction and which is blocked upon Porcn inhibition (LGK974), in Porcn$^{KO}$ cells and by using a palmitoylation-deficient S209A Wnt3A mutant.

B  Wild-type or stable Wnt3-and Wnt5B-expressing HEK293T cells were treated with 5 μM LGK974 for 48 h and subjected to Western blot analysis.

C  Western blot analysis of endogenous Evi in wt, Porcn$^{KO}$, or Evi$^{KO}$ HEK293T cells upon overexpression of Wnt3A or IGFBP5-V5. Porcn$^{KO1.2}$ and Porcn$^{KO1.4}$ indicate clone #2 and clone #4 of Porcn$^{KO}$ HEK293T cells generated with Porcn sgRNA1 (Appendix Fig S3). Clonal Evi$^{KO}$ HEK293T cells were generated with Evi sgRNA2 (Evi$^{KO2.9}$; clone #9) or Evi sgRNA1 (Evi$^{KO1.1}$; clone #1; Appendix Fig S2). Increase in total β-catenin protein served as control for Wnt pathway activation.

D  Western blot analysis of endogenous Evi in HEK293T cells transfected with the indicated overexpression plasmids. When indicated, the cells were additionally treated with 5 μM LGK974 for 48 h.

E  Western blot analysis of endogenous Evi in HCT116 or A375 cells treated with 5 μM LGK974 or DMSO for the indicated hours (h). All Western blots are representative of three independent experiments. β-Actin was used as a loading control, LRP6 as a reference membrane protein and EGF-Myc and IGFBP5-V5 as controls for secreted proteins. Specific Evi bands are indicated by arrows, and unspecific bands are marked by asterisks.

F  Scheme: Wnt-induced Evi stabilization is blocked in the absence of Wnt palmitoylation (Porcn$^{KO}$, LGK974, Wnt3A S209A).

Source data are available online for this figure.

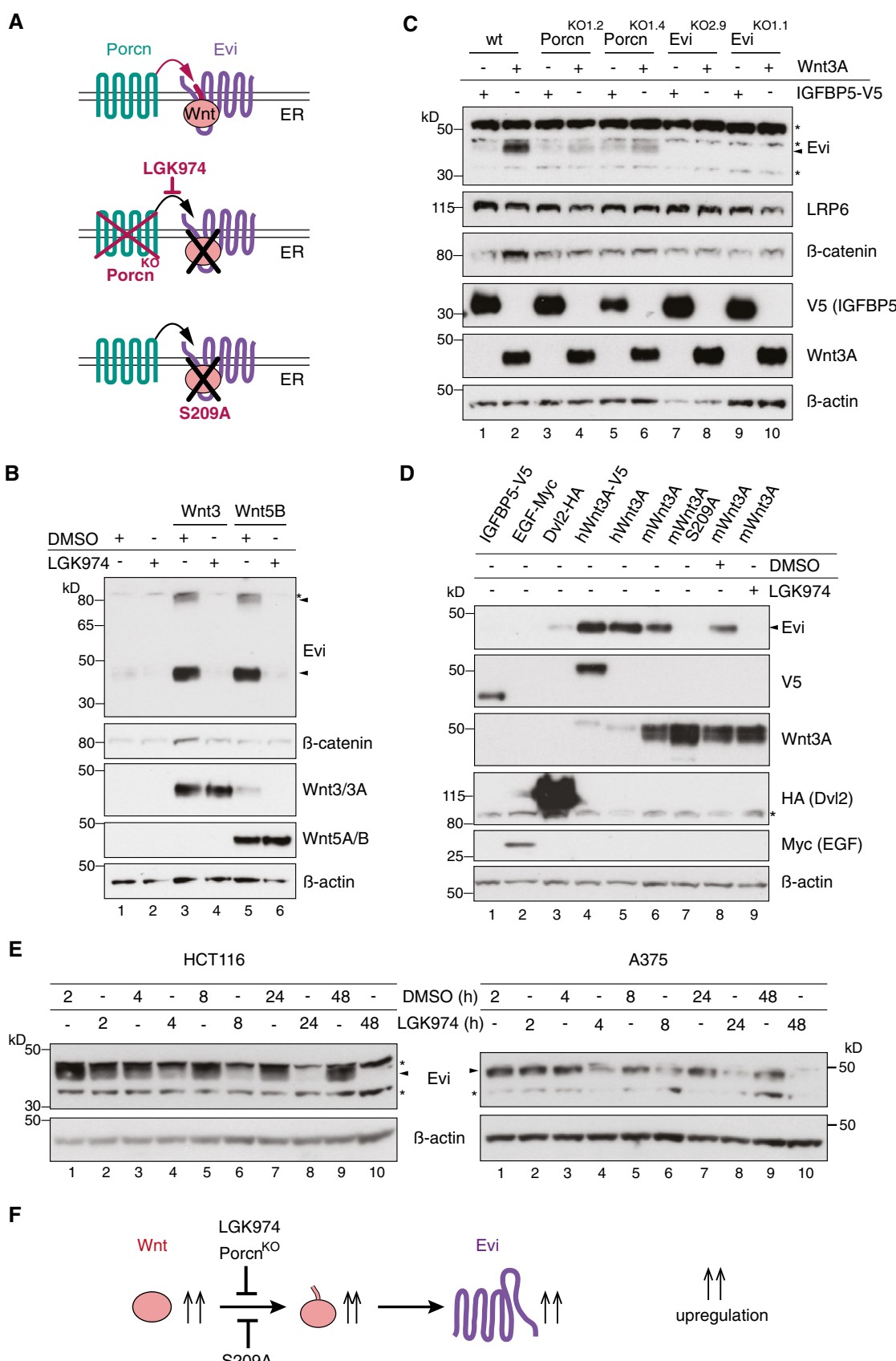

**Figure 2.**

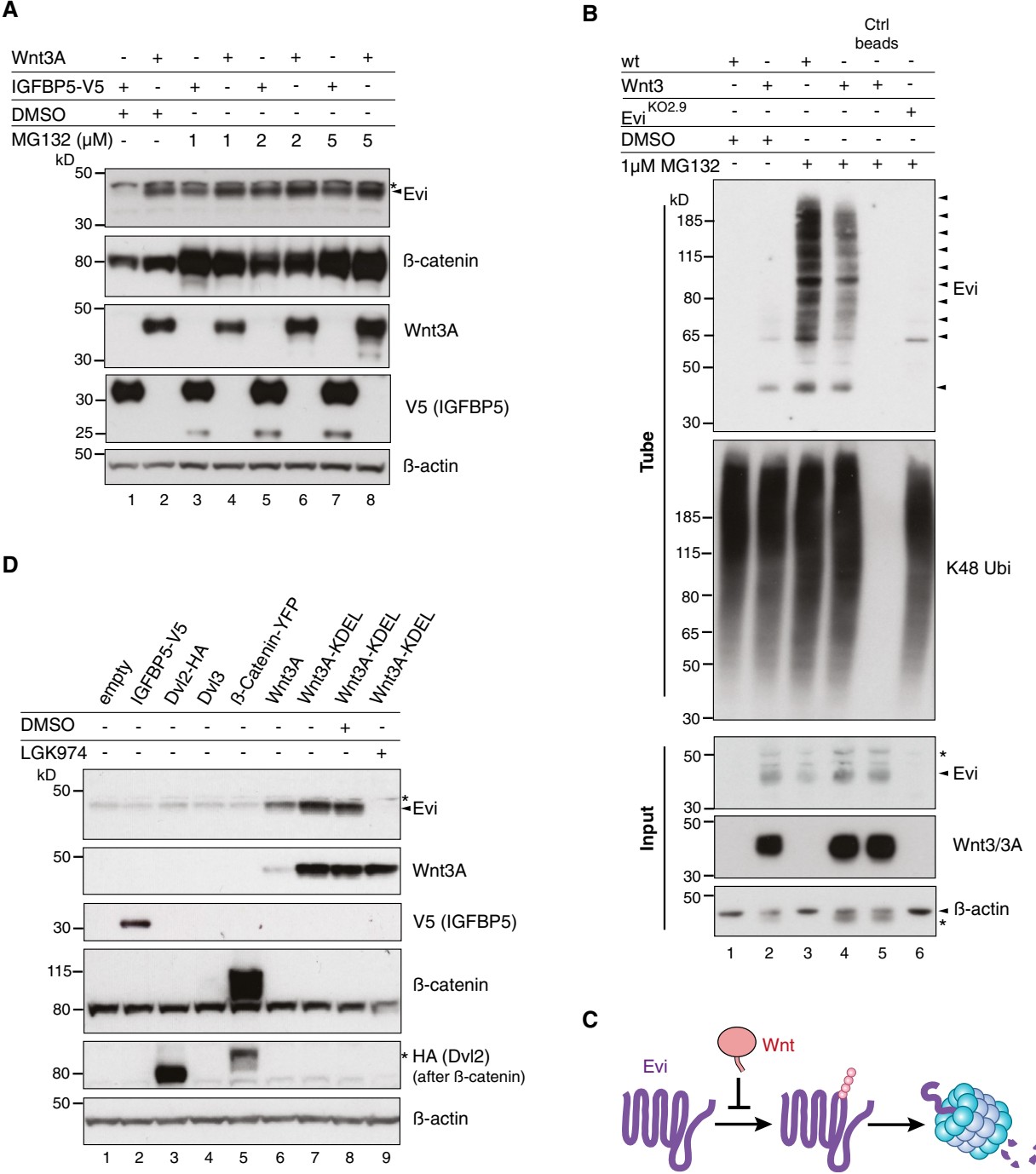

**Figure 3. Evi poly-ubiquitination is regulated by the presence of Wnt proteins.**

A   HEK293T cells were transfected with Wnt3A or IGFBP5-V5 expression constructs and treated with MG132 at the indicated concentrations for 24 h. Cell lysates were analyzed for endogenous Evi by immunoblotting. Total β-catenin protein was used to assess MG132 efficiency.

B   Wild-type (wt), stable Wnt3-expressing, or Evi$^{KO2.9}$ HEK293T cells were treated with 1 μM MG132 for 24 h. TUBE2 immunoprecipitates were assayed for endogenous Evi or K48 poly-ubiquitin by immunoblotting. To confirm specificity of the TUBE2 assay, Ctrl agarose beads were used as control. The asterisk at the β-actin blot indicates Wnt3A proteins blotted before membrane stripping.

C   Scheme illustrating ubiquitination and proteasomal degradation of Evi, which is blocked in the presence of Wnt ligands.

D   HEK293T cells were transfected with the indicated plasmids and additionally treated with 5 μM LGK974 for 48 h when indicated. In case of Wnt3A-KDEL, the ER-retaining sequence KDEL was C-terminally fused to Wnt3A. Dvl2-HA, Dvl3, and β-catenin-YFP overexpression was used as negative control to verify that Evi stabilization was not due to downstream activation of Wnt signaling. All Western blots are representative of three independent experiments. β-Actin was used as loading control. Specific Evi bands are marked by arrows and unspecific bands by asterisks.

Source data are available online for this figure.

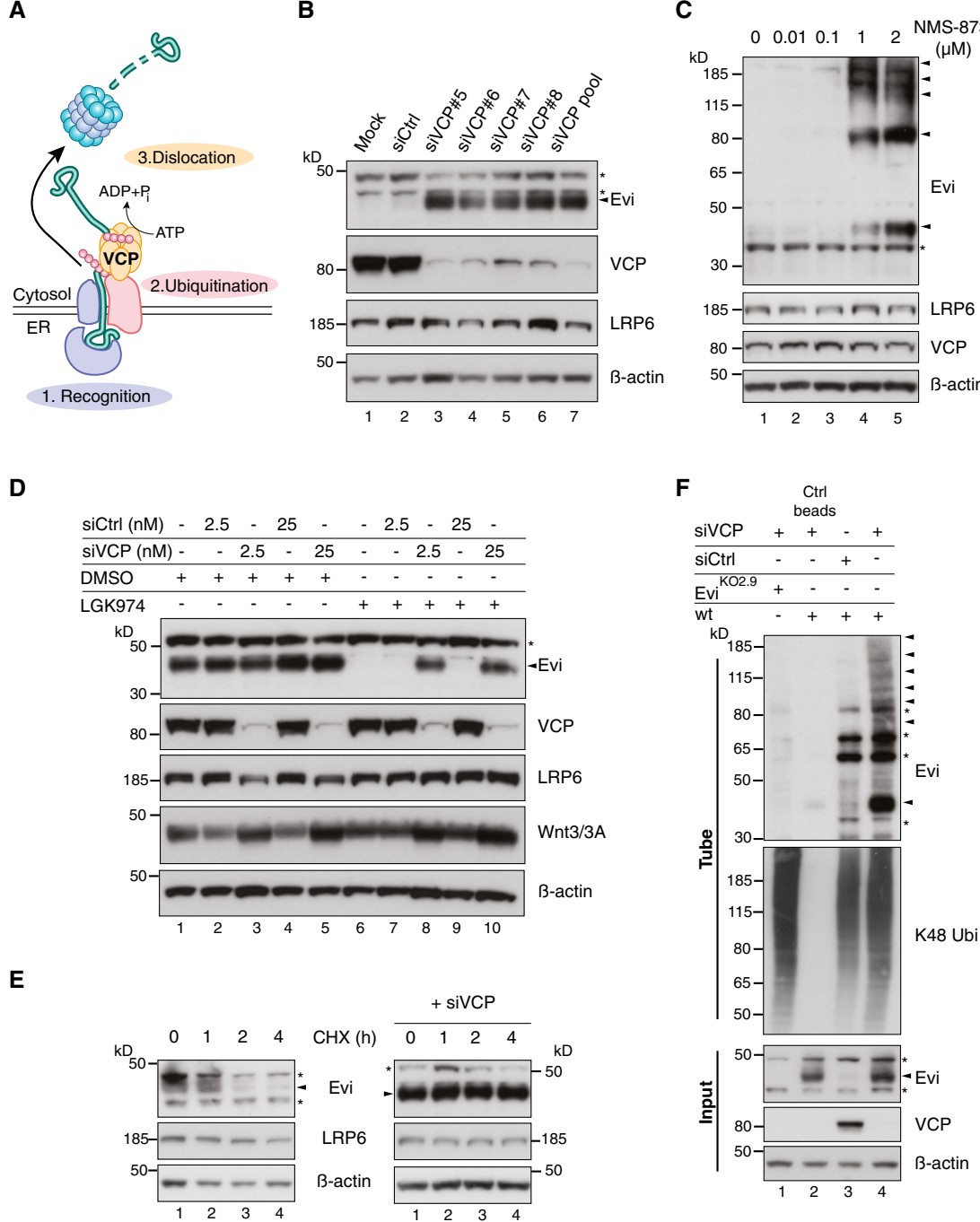

**Figure 4.  The ERAD-protein VCP is required for Evi destabilization.**

A   ERAD-dependent degradation requires recognition, ubiquitination, and subsequent dislocation of the selected substrates through the ER membrane to cytosolic proteasomes. ATP-dependent dislocation is mediated through the AAA-type ATPase VCP.

B   Western blot analysis of endogenous Evi and the indicated proteins in HEK293T cells upon knockdown of VCP using single or pooled VCP siRNAs.

C   HEK293T cells were treated for 24 h with the VCP inhibitor NMS-873 at the indicated concentrations and analyzed for the indicated proteins by immunoblotting.

D   Following reverse transfection with pooled VCP or Ctrl siRNAs (2.5 nM or 25 nM), stable Wnt3-expressing HEK293T cells were treated with 5 μM LGK974 or DMSO for 48 h and analyzed for the indicated proteins.

E   After reverse transfection with pooled VCP siRNAs (if indicated), HEK293T cells were treated with 50 μM cycloheximide (CHX) for the indicated hours (h). Cell lysates were analyzed by immunoblotting.

F   Evi^KO2.9 or wild-type (wt) HEK293T cells were reverse transfected with pooled VCP or Ctrl siRNAs and analyzed for ubiquitinated Evi and K48 ubiquitinated proteins by TUBE2 precipitation. All Western blots are representative of three independent experiments. β-Actin was used as a loading control and LRP6 as a reference membrane protein involved in Wnt signaling. Specific Evi bands are indicated by arrows and unspecific bands by asterisks.

Source data are available online for this figure.

We demonstrated above that Evi protein levels are reduced upon Porcn inhibition. To determine whether VCP is involved in Evi degradation when Porcn activity is absent, we analyzed the combinatorial effect of VCP knockdown and LGK974 treatment in Wnt3-expressing HEK293T cells. Notably, VCP knockdown rescued Evi degradation upon Porcn inhibition (Fig 4D, lanes 8 and 10) without affecting Evi protein levels in DMSO-treated HEK293T-Wnt3 cells (Fig 4D, lanes 3 and 5). VCP-dependent degradation of Evi in Wnt-expressing cells thus only occurred upon Porcn inhibition indicating that the absence of lipid-modified Wnt proteins triggered Evi degradation via VCP. These steady-state results were extended by monitoring turnover rates of endogenous Evi in the presence and absence of VCP activity. Evi was turned over by 4 h of cycloheximide treatment in HEK293T cells, but remained stable upon VCP knockdown (Fig 4E). In addition, VCP knockdown caused an accumulation of ubiquitinated Evi indicating that poly-ubiquitination occurred prior to its engagement with VCP (Fig 4F). The high molecular weight bands attributed to poly-ubiquitinated forms were specific for Evi as they were not observed in Evi$^{KO}$ HEK293T cells. These results indicate that endogenous Evi is turned over via ERAD in a VCP-dependent process in the absence of mature Wnt proteins.

**Endogenous Evi interacts with Porcn and VCP**

Since Wnt-induced Evi stabilization takes place in the ER (Fig 3D), a Wnt-Evi interaction might occur in this compartment, as has been suggested previously (Yu *et al*, 2014). Porcn is an ER-resident protein and responsible for Wnt palmitoylation, which is a pre-requisite for the subsequent binding of Wnt proteins to Evi (Herr & Basler, 2012). We asked whether Porcn might interact with Evi to facilitate direct transfer of palmitoylated Wnts. To investigate a potential Evi-Porcn interaction, FLAG-tagged Porcn was co-expressed with either Wnt3A or IGFBP5 and Porcn-interacting proteins were co-immunoprecipitated. In the presence of Wnt3A, Porcn bound to endogenous Evi (Fig 5A, lane 2), but not to N-cadherin, an unrelated integral membrane protein (Fig EV4A). When IGFBP5-V5 was expressed instead of Wnt3A, we observed a reduction of the Evi-Porcn interaction (Fig 5A, lane 3), which could be linked to reduced Evi stability in the absence of Wnt proteins (Fig 5A, Input lane 3). Notably, blocking Evi degradation through VCP knockdown resulted in comparable amounts of Evi bound to Porcn in the presence or absence of Wnt3A (Fig 5A, lanes 5 and 6). In addition, higher molecular weight forms of Evi bound to Porcn upon VCP knockdown, indicating that poly-ubiquitinated Evi could still engage Porcn. Together, these findings show that the interaction between endogenous Evi and Porcn does not depend on Wnt proteins once Evi degradation is blocked.

The abundance of Evi was strongly dependent on VCP activity (Fig 4B and C), suggesting that endogenous Evi might interact with VCP. Since interactions between VCP and substrates are often transient and difficult to detect, we used a catalytically dead mutant form of VCP (VCP-DKO) in addition to wild-type VCP. With mutations in both ATPase domains (E305Q/E578Q), VCP-DKO cannot process and thus traps its substrates (Dalal *et al*, 2004; Tresse *et al*, 2010). Pull-down analysis of the GFP-tagged VCP variants revealed that both unmodified and ubiquitinated Evi interacted with catalytically inactive VCP-DKO (Fig 5B, lanes 5 and 6), affirming that Evi is a VCP client. In contrast to Evi, N-cadherin did not interact with VCP-DKO, confirming specificity of the Evi-VCP-DKO interaction (Fig EV4B). VCP-DKO forms hexamers with endogenous VCP and thereby disrupts endogenous VCP activity (Tresse *et al*, 2010). Consistently, the ubiquitination and steady-state level of Evi were increased upon overexpression of VCP-DKO (Fig 5B, lanes 5 and 6), further providing evidence that Evi is a VCP client. To a lesser extent, we also detected binding of Evi to wild-type VCP upon Wnt3A overexpression (Fig 5B, lane 4), which was not detectable in IGFBP5 controls. However, once Evi degradation was prevented by the dominant negative VCP-DKO, the interaction between Evi and VCP-DKO occurred independently of Wnt proteins. Together, these results suggest that the binding of Evi to both Porcn and VCP occurs independent of Wnt proteins when Evi degradation is prevented.

This result was surprising, since we expected to observe more binding of Evi to Porcn in the presence of Wnt proteins to support Wnt secretion and an increased interaction with VCP in the absence of Wnts to promote Evi degradation. One possible explanation was that Evi might interact simultaneously with Porcn and VCP independent of Wnt ligands. In this case, the fate of Evi would be determined within one triaging complex, resulting in either stabilization of Evi in the presence of Wnt proteins or Evi degradation through ERAD if Evi is not needed.

To explore the presence of a complex consisting of Evi, Porcn, and VCP, we performed sequential immunoprecipitations. Following immunoprecipitating FLAG-tagged Porcn, the FLAG peptide eluates were used to pull down GFP-tagged VCP-DKO. Probing the two-step immunoprecipitates for Evi revealed bands corresponding to unmodified and potentially ubiquitinated Evi, indicating that endogenous Evi was bound to VCP and Porcn simultaneously (Fig 5C, double IP). These results support a model whereby Evi forms a complex in the ER membrane with both Porcn and the VCP-containing ERAD machinery. Since the Porcn-VCP interaction occurred independently of Evi in wild-type and Evi$^{KO}$ HEK293T cells (Fig EV4C), a pre-existing Porcn-VCP complex might function as a "triage platform" for newly translated Evi to directly channel Evi either into ERAD or the secretory pathway (Fig EV4D), responding to the need of Wnt protein secretion.

---

**Figure 5. Evi forms a complex with Porcn and VCP simultaneously.**

A    After VCP knockdown (if indicated) and expression of the indicated proteins in wild-type (wt) or Evi$^{KO}$ HEK293T, a FLAG IP was performed and analyzed by Western blotting.

B, C    Wild-type (wt) or Evi$^{KO}$ HEK293T cells were transiently transfected with the indicated overexpression constructs and treated with 1.5 μM NMS-873 for 24 h. (B) GFP co-immunoprecipitation of GFP-tagged wt VCP and VCP-DKO was followed by Western blot analysis. (C) Following FLAG IP, interacting proteins were eluted from the beads (FLAG IP) and subjected to GFP pull-down (double IP) and analyzed by immunoblotting. All Western blots are representative of three independent experiments. β-Actin served as loading control. Specific Evi bands are indicated by arrows and unspecific bands by asterisks.

Source data are available online for this figure.

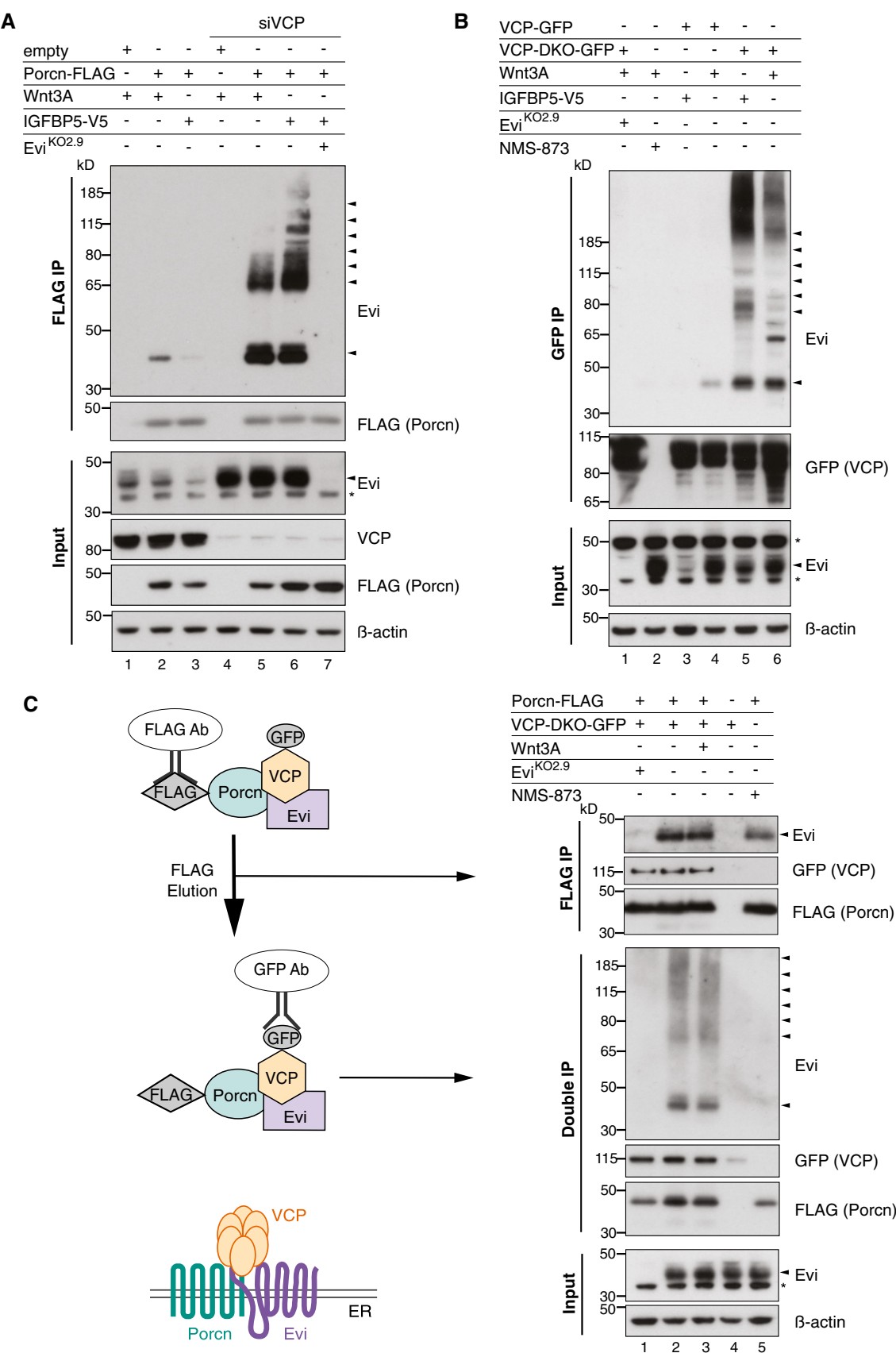

**Figure 5.**

## UBE2J2 and the ubiquitin E3 ligase CGRRF1 are involved in Evi regulation

The detection of poly-ubiquitinated Evi and its proteasome-dependent degradation prompted us to identify the responsible ubiquitination machinery. For this purpose, we knocked down the ubiquitin-conjugating enzymes (E2s) UBE2J1, UBE2J2, and UBE2G2, which have been previously linked to ERAD processing (Lenk *et al*, 2002; Christianson *et al*, 2011; van den Boomen *et al*, 2014) and included UBE2U as an ERAD-independent E2 enzyme. Silencing UBE2J2 with either pooled (Fig 6B) or individual siRNAs (Fig EV5B) increased Evi protein levels that were comparable to those observed with VCP knockdown (Fig 6B). By contrast, silencing of neither UBE2U nor the other ERAD E2 enzymes stabilized Evi. Furthermore, UBE2J2 knockdown reduced the appearance of poly-ubiquitinated Evi (Fig 6D). These results support the participation of UBE2J2 in the ubiquitination and degradation of Evi.

UBE2J2 is tethered to the ER membrane by a C-terminal tail anchor (Fig 6A), suggesting that a corresponding ubiquitin ligase (E3) may also reside in the ER. To identify E3 ligases involved in Evi degradation, we used siRNAs to screen a selected panel of E3s implicated previously in ERAD or predicted to reside in the ER membrane (Neutzner *et al*, 2011; Figs 6C and EV5A). Unexpectedly, depletion of E3s with well-characterized roles in ERAD such as Hrd1 (Hampton *et al*, 1996) and gp78 (Fang *et al*, 2001) was not sufficient to stabilize Evi. Instead, Evi was stabilized upon knockdown of CGRRF1, a relatively uncharacterized ER-resident E3 ligase that has not yet been linked to ERAD (Figs 6C and EV5C). Like UBE2J2, knockdown of CGRRF1 reduced Evi ubiquitination (Fig 6D). Residual poly-ubiquitinated Evi upon CGRRF1 and UBE2J2 knockdown indicate that additional E2 and E3 enzymes might contribute to the poly-ubiquitination of Evi. To further confirm the involvement of CGRRF1 in Evi processing, a potential interaction between CGRRF1 and Evi was investigated. Indeed, endogenous Evi co-precipitated with stably expressed FLAG-HA-tagged CGRRF1 but not with Hrd1 (Fig 6E). *In silico* analysis of CGRRF1 predicts a transmembrane domain near its N-terminus and a cytosolic RING domain ($C_3HC_4$) at its C-terminus (Neutzner *et al*, 2011; Fig 6A). Mutation of the 2nd and 3rd (C2/3A) or 2nd and 4th (C2/4A) cysteine residues within the RING domain of CGRRF1 (Fig 6A) was sufficient to stabilize the interaction between endogenous Evi and CGRRF1 (Fig EV5D, FLAG IP). Expression of these CGRRF1 RING mutants additionally increased Evi steady-state abundance (Fig EV5D, Input) indicating that the

RING domain of CGRRF1 is required for Evi degradation. Since both UBE2J2 and CGRRF1 influenced the abundance and poly-ubiquitination of Evi, we hypothesized that they could be part of one ubiquitination complex. Indeed, we could detect an interaction between V5-tagged UBE2J2 and FLAG-HA-tagged CGRRF1 (Fig EV5E). Taken together, our data indicate that CGRRF1 and UBE2J2 are part of an ubiquitination complex that can regulate Evi abundance (Fig EV5F). CGRRF1 expression is reduced in colon and endometrial cancer (Fig EV5G; TCGA, 2012), which correlates with increased Evi protein levels (Voloshanenko *et al*, 2013; Stewart *et al*, 2015) and a lack of transcriptional upregulation of Evi as assessed by TCGA expression data (Fig EV5H; TCGA, 2012).

## Evi stabilization increases Wnt secretion

Since Evi appears to be continuously degraded by ERAD, we next asked whether Wnt secretion would increase if Evi degradation is blocked through VCP knockdown. We used an experimental setup of short, transient expressed Wnt ligands to avoid Evi stabilization through continuous high Wnt levels. When Wnt secretion was monitored shortly after Wnt expression but before sufficient Evi stabilization by Wnts, VCP knockdown elevated the secretion of Wnt5A and Wnt3A (Fig 7A and B; Appendix Fig S4A). Similarly, precipitating secreted Wnt3A ligands upon knockdown of either UBE2J2 or CGRRF1 displayed an increase in Wnt secretion (Fig 7C; Appendix Fig S4B). Enhanced Wnt secretion upon knockdown of VCP, UBE2J2, or CGRRF1 co-occurred with increased Evi levels in cell lysates (Fig 7A–C), indicating that the stabilized Evi protein is functionally active and able to support the secretion of Wnt ligands. These findings indicate a regulatory role of the ERAD pathway, the E2 conjugating enzyme UBE2J2, and the E3 ligase CGRRF1 in Evi protein turnover and with an impact on Wnt protein secretion (Fig 7D).

## Discussion

Within the early secretory pathway, ERAD ensures the degradation of misfolded and misassembled proteins (Vembar & Brodsky, 2008). Besides its role in ensuring protein quality, ERAD also has the capability to respond to the physiological requirement of mature proteins that transit or reside within the ER to regulate their homeostasis accordingly. This is well described for the negative feedback regulation of sterol biosynthesis, where abundant sterol metabolites

---

**Figure 6. UBE2J2 and CGRRF1 are involved in the degradation and ubiquitination of Evi.**

A Schematic illustration of UBE2J2 and CGRRF1. The transmembrane (TM) domain of CGRRF1 was predicted using TMHMM server v. 2.0 (Krogh *et al*, 2001). The core catalytic domain of the E2 conjugating enzyme UBE2J2 (amino acid residues 14–127) is depicted as UBCc. Amino acid residues important for the interaction with E3 ligases are shown in green, residues facing the ubiquitin thioester interaction side in blue, and the active site cysteine in red. The catalytic zinc finger domain of CGRRF1 consists of the typical $C_3HC_4$ RING type motif and spans the amino acid residues 274–315. Active site cysteine residues are marked in red and the active site histidine in orange.

B, C Following reverse transfection of HEK293T cells with pooled siRNA against the indicated proteins, the cell lysates were analyzed by immunoblotting.

D Following UBE2J2 and CGRRF1 knockdown, HEK293T were treated with 1 μM MG132 for 24 h. Poly-ubiquitinated proteins were TUBE2 precipitated and stained for endogenous Evi or K48 poly-ubiquitin. Ctrl agarose beads were used to confirm specifity of the TUBE2 assay for ubiquitinated proteins.

E CGRRF1-FLAG-HA (FH) and Hrd1-FLAG-HA (FH) inducible Flp-IN HEK293T cells were treated with 1 ng/ml doxycycline for 18 h and 10 μM MG132 for 4 h prior to FLAG IP and subsequent immunoblotting with mouse monoclonal Evi antibody (clone YJ5). All Western blots are representative of three independent experiments. β-Actin was used as loading control. Specific Evi bands are indicated by arrows and unspecific bands by asterisks.

Source data are available online for this figure.

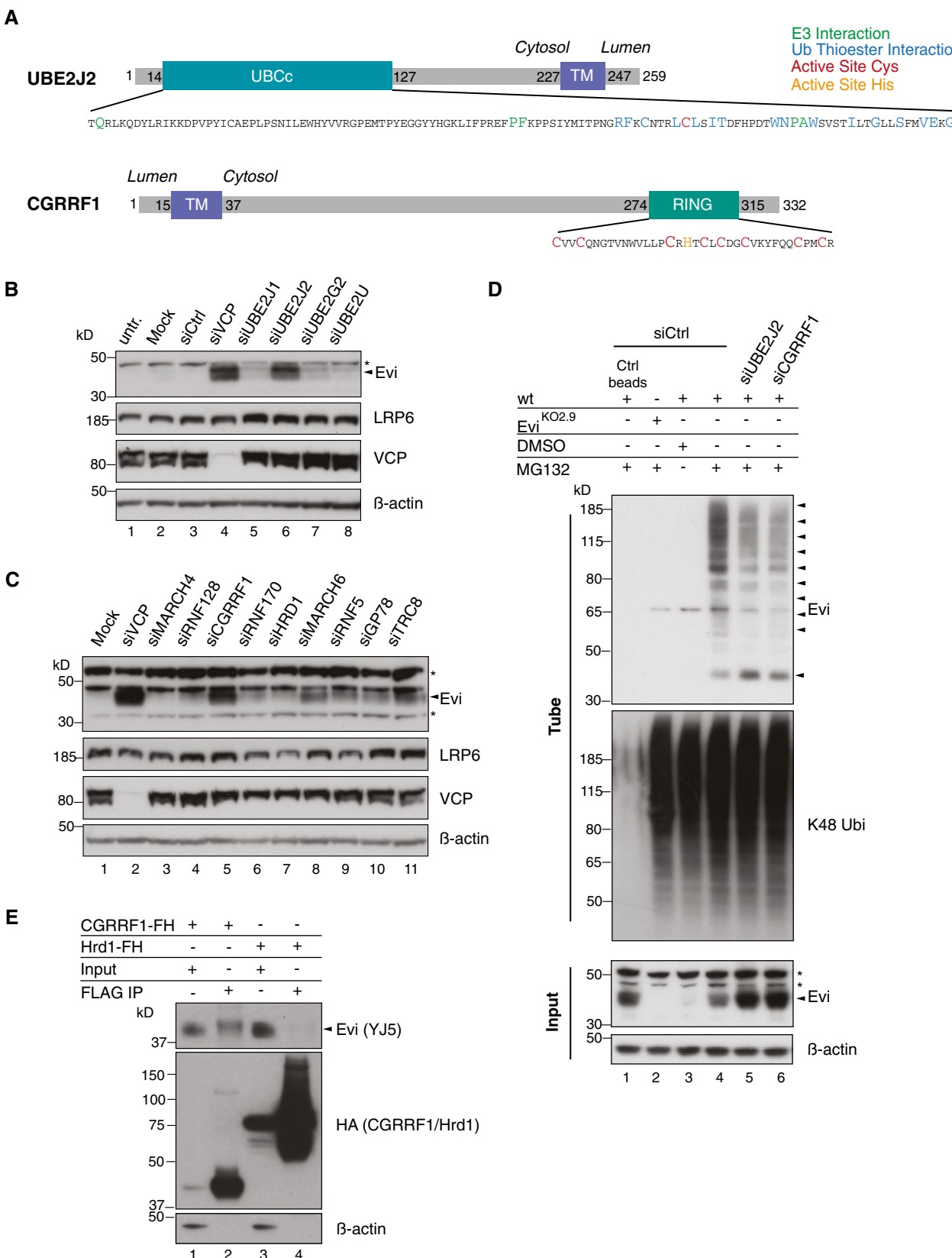

**Figure 6.**

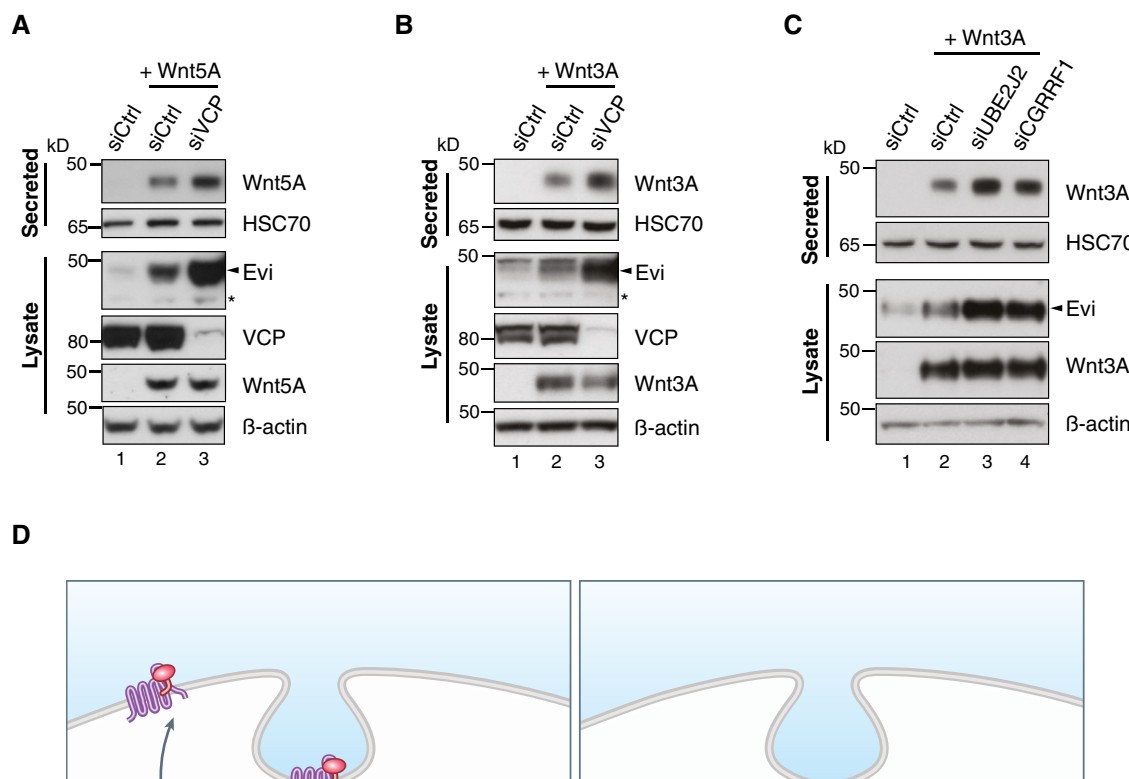

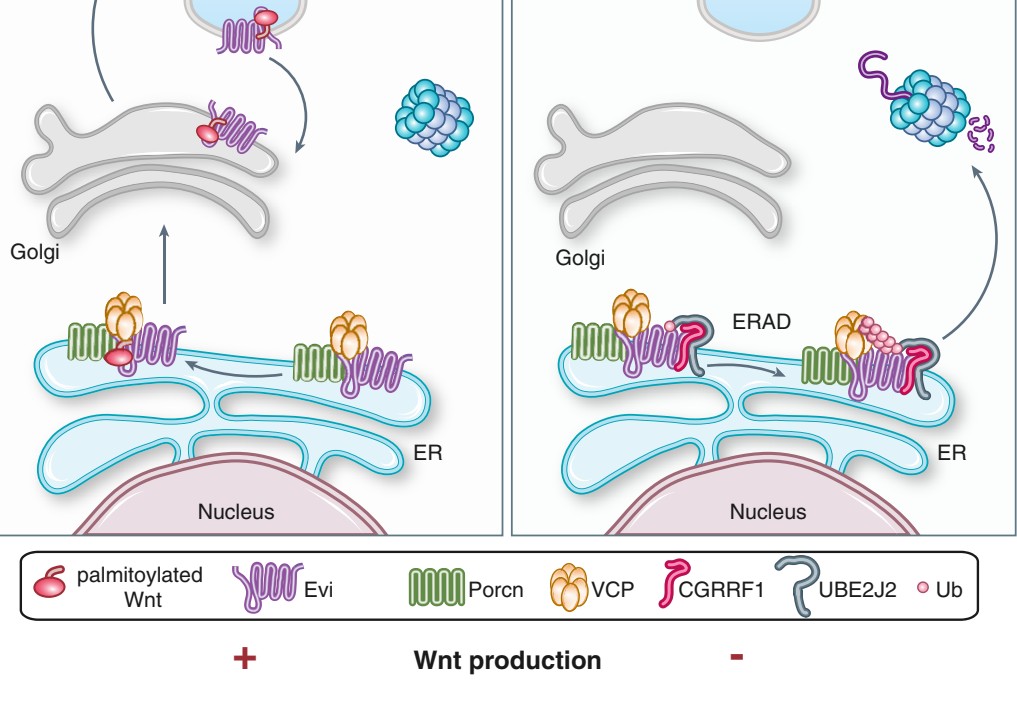

**Figure 7. Wnt secretion is modulated by ERAD.**

A–C   Twenty-four hours after reverse transfection with Ctrl siRNA or pooled siRNAs against VCP, UBE2J2, or CGRRF1, HEK293T cells were transfected with (A) Wnt5A or (B, C) Wnt3A expression plasmids. Using Blue Sepharose, secreted Wnt proteins were precipitated from conditioned medium 24 h after plasmid transfection. HSC70 also binds to Blue Sepharose and served as loading control. Western blot quantification of three (B) and six (C) replicates is shown in Appendix Fig S4A and B.

D      Model of ERAD-dependent regulation of Wnt protein secretion. In the absence of Wnt proteins (− Wnt production), Evi is poly-ubiquitinated and degraded via ERAD, which involves the E3 ligase CGRRF1 and the E2 conjugating enzyme UBE2J2. Poly-ubiquitinated Evi is translocated via the ATPase VCP into the cytosol and degraded by the proteasome. In the presence of mature Wnt proteins (+ Wnt production), Evi is directed through Porcn into the Wnt secretory pathway and not degraded via ERAD. Stabilized Evi is capable to cope with the need of increased Wnt protein secretion. Having reached the plasma membrane, Evi can be recycled back to the Golgi and ER and Wnt proteins are secreted directly or after endocytosis through exovesicles. Despite the existence of a complex containing Evi, Porcn, and VCP, Porcn and VCP do not necessarily need each other to exert their functionality.

Source data are available online for this figure.

induce ERAD-mediated degradation of sterol biosynthetic enzymes like 3-hydroxy-3-methylglutaryl-coenzyme-A reductase (HMGCR) (Gil *et al*, 1985; Sever *et al*, 2003) and squalene monooxygenase (SQLE) (Gill *et al*, 2011; Foresti *et al*, 2013). Beyond this pathway, only a few other substrates have been identified for regulatory ERAD. Given the wide range of processes housed at the ER, there are likely more physiological events that are regulated through ERAD. Moreover, the sensing and recruiting mechanisms along with the participation of different ubiquitination machineries offer another layer of complexity that remains poorly understood. Identifying endogenous substrates and regulatory components are thus of great interest to better understand the physiological scope of regulatory ERAD (Ruggiano *et al*, 2014).

We provide several lines of evidence for a novel role of ERAD in the homeostatic regulation of the Wnt cargo receptor Evi (Fig 7D). We show that (i) palmitoylated Wnt ligands stabilize Evi within the ER in a Porcn-dependent manner, and (ii) in the absence of Wnt ligands, Evi is ubiquitinated involving the E2 enzyme UBE2J2 and the E3 ligase CGRRF1 and (iii) constitutively degraded by a VCP and proteasome-dependent process, which (iv) is decided by a triaging complex containing Porcn and VCP that directs Evi either into ERAD or the secretory pathway. Depending on the need of Wnt transportation, such triaging complex would allow the channeling of Evi either into the secretory pathway through Porcn or into ERAD through VCP. Despite their coordinated participation in Evi regulation, VCP and Porcn do not necessarily require each other for their functionality: VCP still induced the degradation of Evi in the absence of Porcn activity (Fig 4D), and Porcn could guide Evi into the Wnt secretory pathway upon VCP knockdown (Fig 7A and B). With Porcn, VCP, UBE2J2, and CGRRF1, we identified novel factors involved in the regulation of endogenous Evi in response to the physiological need to export Wnt ligands. When Wnt expression is induced, cells must have an adequate capacity within the Wnt secretory machinery to accommodate changes in demand. We show that cells meet this requirement through a rapid, positive feedback mechanism based on Evi stabilization. While the cargo receptor Evi is continuously degraded through ERAD in the absence of Wnt ligands, an increase in Wnt protein levels stabilizes Evi by precluding its ERAD-mediated degradation. Elevated Evi protein levels, in turn, increase the cell's capacity to transport Wnt proteins to the cell surface (Fig 7D).

Secreted Wnt proteins can act as morphogens and are implicated in many processes during embryonic development and adult tissue homeostasis (Logan & Nusse, 2004). In addition, de-regulation of the Wnt pathway has been implicated in the onset and progression of tumorigenesis (Clevers, 2006). A better understanding of the mechanisms underlying Wnt signaling therefore might lay the foundation for novel therapeutic approaches. Blocking Wnt secretion with the Porcn inhibitor LGK974 induced regression of multiple tumor models *in vivo* (Liu *et al*, 2013) and is currently being investigated in clinical studies (Lum & Chen, 2015; Zhan *et al*, 2017). Despite the high relevance of secreted Wnt ligands in different cancers and the reported additional Wnt-independent functions of Porcn (Covey *et al*, 2012; Erlenhardt *et al*, 2016), inhibition of Porcn is to date the only pharmacological approach to inhibit Wnt secretion.

Evi is an essential component of the Wnt secretion machinery (Bänziger *et al*, 2006; Bartscherer *et al*, 2006; Goodman *et al*, 2006), and its protein levels are increased in cancer compared to normal colon tissue (Voloshanenko *et al*, 2013; Stewart *et al*, 2015).

Since Evi must mediate the secretion of different Wnt proteins, it must be available as soon as any of the Wnt proteins is expressed. Transcriptional co-regulation with 19 different Wnt genes during developmental and homeostatic processes would require a complex transcriptional control mechanism, which could explain why Evi is continuously produced but degraded in the absence of Wnt proteins. Moreover, ERAD-dependent regulation of Evi at the post-translational level enables the cells to rapidly respond and accommodate changes in Wnt expression.

Our data indicate that Porcn could not only be important as a palmitoyl-transferase, but could additionally function as an "ERAD-triaging protein" during Wnt protein secretion. The role of Porcn in ERAD-dependent control of Wnt secretion could be analogous to that of microsomal triglyceride transfer protein (MTP) in lipid transportation (Brodsky & Fisher, 2008). Similar to the role of MTP in loading lipids onto ApoB, Porcn could be required to transfer mature Wnt proteins onto Evi. The absence of Porcn or MTP as well as the absence of the Wnt or lipid cargo induces the degradation of the corresponding cargo shuttling receptor. In contrast, successful loading of Wnts onto Evi or lipids onto ApoB in the presence of Porcn or MTP permits ER exit and trafficking of the loaded carriers to the Golgi. Porcn and MTP could thus be substrate-specific ERAD regulators of either Evi or ApoB to modulate the secretion of Wnts and lipids, respectively, via an ERAD-dependent positive feedback mechanism.

It is tempting to speculate that Porcn could "moonlight" as an ERAD-triaging protein in the secretion of proteins other than Wnts. Recently, Porcn has been shown to interact with AMPA receptors and to regulate their abundance (Erlenhardt *et al*, 2016). Porcn may thus assist with the assembly of protein complexes in the ER such as an Evi-Wnt complex or the multimeric AMPA receptor complex. Since such complex assembly would only be possible when all subunits are present, Porcn could support either the secretion of the fully assembled complex or the degradation of the remaining subunits. Since Evi appears to interact with Porcn, continuous degradation of "unused" Evi could ensure that additional functions of Porcn are not blocked by the bound Evi protein.

The covalent modification of proteins by ubiquitin requires the coordinated activity of the E1-E2-E3 enzymes as part of the ubiquitination cascade (Dikic *et al*, 2009). We showed that the E2-conjugating enzyme UBE2J2 and the largely uncharacterized E3 ligase CGRRF1 affect the poly-ubiquitination and degradation of Evi. E3 ligases confer specificity of ubiquitin conjugation by interacting with the substrates to promote efficient ubiquitin transfer. While some ER-resident E3s like Hrd1 (Hampton *et al*, 1996; Bernasconi *et al*, 2010; Kanehara *et al*, 2010; Christianson *et al*, 2011) seem to accommodate a broad range of substrates, others appear to be more selective including TRC8 (Stagg *et al*, 2009) and RNF170 (Lu *et al*, 2011). We identified the RING-finger protein CGRRF1 as an Evi-binding E3 ligase that modulates the abundance of Evi. Little is known about CGRRF1, but a previous study has proposed that CGRRF1 is an inactive E3 ligase (Kaneko *et al*, 2016). This study, however, paired CGRRF1 with the promiscuous E2 UbcH5a for *in vitro* assays, which might suggest that CGRRF1 is only functionally active with particular E2 enzymes such as UBE2J2. The demonstration that depletion of CGRRF1 and UBE2J2 reduced Evi ubiquitination and increased Evi steady-state levels indicate an active role for a CGRRF1-UBE2J2 complex in Evi regulation.

Notably, CGRRF1 mRNA levels are reduced in several cancers including endometrial and colon adenocarcinomas (Fig EV5G, TCGA). Reduced CGRFF1 levels and increased expression of Wnt ligands (Fig EV1A) could explain the cancer-correlated high Evi protein levels in the absence of elevated Evi transcription (Fig EV5H; TCGA, 2012). Thus, the Wnt- and ERAD-dependent mechanism of Evi regulation could be a novel starting point to target Wnt driven malignancies.

Our study identified Porcn, VCP, UBE2J2, and the largely uncharacterized E3 ligase CGRRF1 as new factors regulating Evi homeostasis. In addition, our findings describe an adaptive ERAD pathway in mammalian cells that controls a protein secretion event by adjusting the abundance of a key cargo receptor to the level of its ligands. ERAD-mediated homeostatic control of proteins could also be involved in other secretory pathways, as many secreted and signaling proteins are produced within the ER and their access to the extracellular space must be tightly controlled.

# Materials and Methods

## Patient FFPE tissue

Formalin-fixed paraffin-embedded (FFPE) tissue sections from healthy colon and matched colon adenocarcinoma (stage G2) were obtained from the tissue bank of the National Center for Tumor Diseases (NCT, Heidelberg, Germany) in accordance with the regulations of the tissue bank and the approval of the ethics committee of Heidelberg University (Ethics vote 206/2005).

## Cell lines

HEK293T (ATCC, #CRL-3216) and A375 (ATCC, #CRL-1619) cells were cultured in DMEM (Gibco) and HCT116 cells (ATCC, #CCL-247) in McCoy's (Gibco) medium without antibiotics at 37°C and 5% $CO_2$ in a humidified atmosphere. All media were supplemented with 10% fetal bovine serum (Biochrom), and all cells were regularly confirmed to be mycoplasma negative.

## Generation of knockout cells

CRISPR/Cas9 knockout cell lines were generated according to protocols described before in Ran *et al* (2013). In brief, 20-nt single-guide (sg) RNAs were designed using E-CRISP (Heigwer *et al*, 2014). HEK293T cells were transiently transfected in 6-well plates with px459 constructs containing Cas9 and the corresponding sgRNAs and selected with 2 µg/ml puromycin for 48–72 h. Clonal cell lines were generated by limited dilution in 96-well plates. All sequences of sgRNAs are listed in Appendix Table S1.

## Amplicon sequencing

Sequencing was performed essentially as described previously (Voloshanenko *et al*, 2017). CRISPR/Cas9-generated single-cell clones were isolated using DNeasy Blood and Tissue Kit (Qiagen). To detect mutations, primer pairs complementary to 100–150 base pairs up- and downstream of the sgRNA target region were designed using Primer3 database (Koressaar & Remm, 2007). The obtained

primer sequences were fused to adapter sequences for the 2nd PCR. Genomic regions of interest were amplified by PCR and purified using the PCR Cleanup Kit (Machery-Nagel). Illumina adapters were inserted into PCR products using a two-step PCR, and resulting amplicons were sequenced with Illumina MiSeq. Obtained sequences were analyzed by performing multiple sequence alignment using ClustalOmega (Sievers *et al*, 2011). All primer sequences are listed in Appendix Table S1.

## Generation of stable HEK293T cells

For the generation of Wnt3- and Wnt5B-expressing HEK293T cells, lentivirus was produced by transfecting HEK293T cells in 6-well plates with the virus packaging vectors (3 µg psPAX, 2 µg pMDM-VSVG) and 3 µg pLenti-CMV-Wnt3-Blast or pLenti-CMV-Wnt5B-Blast, respectively, using 5 µl TransIT. Forty hours post-transfection, the supernatant was filtered (45 mm), and mixed with DMEM/10% FBS (3:1) and polybrene (2.5 µg/ml). Seeded HEK293T cells were cultured with the virus-containing medium for 2 days, washed with PBS, and selected with 5 µg/ml blasticidin. Stable cell lines were maintained as polyclonal cell populations. Stable, inducible Flp-IN HEK293 cell lines expressing FLAG-HA-tagged Hrd1 or CGRRF1 were generated according to manufacturer's protocols (Invitrogen). Transfection was carried out using Lipofectamine 2000 (Invitrogen) and 1 µg pOG44 Flp-Recombinase Expression Vector (Invitrogen) as well as 1 µg FRT/TO-pcDNA5 encoding FLAG-HA-tagged CGRRF1 or Hrd1. Twenty-four hours after transfection, the medium was changed and cells with integrated pcDNA5-FRT/TO were selected with 100 µg/ml hygromycin. The obtained DOX-inducible cells were treated with 1 ng/ml doxycycline for 18 h to induce sufficient protein expression.

## Quantitative PCR

RNA was reverse-transcribed into complementary DNA by using the RevertAid H Minus First Strand cDNA Synthesis Kit (Thermo Fischer Scientific). 25 ng cDNA was used for quantitative PCR on the Light-cycler 480 (Roche) using the Universal Probe Library system (Roche) in a 384-well format. GAPDH was used as reference gene. Oligonucleotide sequences for quantitative PCR are shown in Appendix Table S1.

## Plasmid transfection

For Evi stability assays, $3 \times 10^5$ cells were transfected in 6-well plates with 700 ng plasmid using 2.8 µl TransIT reagent (VWR). For immunoprecipitation experiments, $3.5 \times 10^6$ cells were transfected in 10-cm dishes with 2 µg of each plasmid (except for 1 µg of Porcn-DDK) using 4 µl TransIT reagent per 1 µg DNA. Cells were harvested 48 h after transfection. The following expression constructs were used: pSpCas9(BB)-2A-Puro (px459; Addgene #48139; Ran *et al*, 2013), psPAX2 (Addgene #12260; D. Trono), pMD2.G-VSVG (Addgene #12259; D. Trono), TCF4/Wnt-Firefly Luciferase (Demir *et al*, 2013), Actin-Renilla Luciferase (Nickles *et al*, 2012), pcDNA3-Wnt11 (Addgene #35922; Najdi *et al*, 2012), pcDNA3-Wnt10B, (Addgene #35921; Najdi *et al*, 2012), pcDNA3-Wnt16 (Addgene #35923; Najdi *et al*, 2012), pcDNA3-Wnt7B (Addgene #35915; Najdi *et al*, 2012), pcDNA3-Wnt8 (Addgene

#35916; Najdi *et al*, 2012), pcDNA3-Wnt5A (Addgene #35911; Najdi *et al*, 2012), pcDNA3-Wnt3A (Addgene #35908; Najdi *et al*, 2012), pcDNA3-Wnt3 (Addgene #35909; Najdi *et al*, 2012), pcDNA3.2-Wnt3A-V5 (Addgene #43810; MacDonald *et al*, 2014), pcDNA3-IGFBP5-V5 (Addgene #11608; S. Johnson), pcDNA3-EGF-Myc (Addgene #11601; Nguyen *et al*, 2000), VCP(wt)-EGFP (Addgene #23971; Tresse *et al*, 2010), VCP(DKO)-EGFP (Addgene #23974; Tresse *et al*, 2010), pCMV6-Myc-DDK-tagged PORCN (Origene # RC223764), sLuc (Katanaev *et al*, 2008), pCS2(+)-mWnt3A (C. Niehrs), pCS2(+)-Dvl3 (C. Niehrs), pCS2(+)-mWnt3A-S209A (Kumar *et al*, 2014), and β-Catenin-YFP (Stemmer *et al*, 2008). To generate C-terminally FLAG-HA-tagged CGRRF1 and Hrd1, the FLAG-HA sequence was inserted into the pcDNA5-FRT/TO vector (Invitrogen). Primers with *HindIII* and *AgeI* restriction site sequences were used to amplify the ORF of Hrd1 (Schulz *et al*, 2017) and CGRRF1 (MGC collection). The extended ORFs of Hrd1 and CGRRF1 were cloned into pcDNA5-FRT/TO-FLAG-HA using *HindIII* and *AgeI* restriction enzymes. UBE2J2 was cloned from pDEST17-UBE2J2 (Addgene #15794; Jin *et al*, 2007) into pcDNA3-V5 vector using in-fusion cloning (Takara Bio Inc, #639642). Wnt3A-KDEL and the RING mutants of CGRRF1-FLAG-HA were generated via site-directed mutagenesis (QuikChange, 200523, Agilent). The primer sequences are listed in Appendix Table S1. For the generation of Wnt3-and Wnt5B stable cells, Wnt3 and Wnt5B were first cloned from pcDNA3-Wnt3 and pcDNA3-Wnt5B into the pDONOR/DEST vector using attB2 primers and subsequently into the pLenti-CMV-Blast-DEST vector (Addgene #17451; Campeau *et al*, 2009).

### RNAi experiments

For 6-well plates, $4 \times 10^5$ HEK293T cells were reverse transfected with 25 nM (if not otherwise indicated) Dharmacon siRNA using 2 μl Lipofectamine RNAiMax reagent (Life Technologies) according to manufacturer's conditions. In 10-cm dishes, knockdown in $3 \times 10^6$ cells was performed using 20 nM Dharmacon siRNA and 8 μl RNAiMax. Cells were harvested 72 h after reverse transfection. Dharmacon identifiers of the used siRNAs are listed in Appendix Table S2.

### Wnt reporter activity assay

Wnt reporter activity assay was performed as described previously (Glaeser *et al*, 2016) with some modifications. Wild-type, Porcn$^{KO1.2}$, and Porcn$^{KO1.4}$ HEK293T cells were transfected in white, flat-bottom 384-well plates with 20 ng TCF4-Firefly Luciferase, 10 ng Actin-Renilla Luciferase and 20 ng of Wnt3A, Dvl3, or pcDNA3.1 plasmids. Dual luciferase readout was performed 72 h after cell seeding using the Mitras LB940 plate reader (Berthold Technologies, Bad Wildbad, Germany). TCF4-Firefly reporter signal was normalized to β-actin-Renilla luciferase signal.

### Inhibitor assays

Cells were treated with 100 ng/ml recombinant mouse Wnt3A (315-20; PeproTech), 10 μM SB216763 (S3442, Sigma), or 50 μg/ml Cycloheximide (C7698, Sigma) for the indicated time points. MG132 (474791, Merck Millipore), Bafilomycin A1 (Cay11038-1, Cayman),

and NMS-873 (5310880001, Merck Millipore) were used for 24 h at the indicated concentrations. Treatment with 5 μM LGK974 (custom synthesis by WuXi AppTec) was performed 6 and 30 h after transfection.

### TUBE2 assay and DUB treatment

For each condition, 40 μl TUBE2 agarose beads (UM402, LifeSensors) were washed three times at RT with TBST and added to 5 mg or 5.5 mg protein lysate. TUBE Ctrl agarose (UM400, LifeSensors) without ubiquitin binding TUBE proteins was included as control to assess unspecific protein binding to the agarose beads. After overnight rotation at 4°C, beads were washed four times with cold TBST at 4°C and taken up in SDS buffer. For DUB treatment, washed TUBE2 beads (with bound poly-ubiquitinated proteins) were incubated for 1 h at 37°C with DUB buffer (50 mM HEPES pH 8.0, 150 mM NaCl, 0.1 mM EDTA, fresh 1 mM DTT), which was supplemented with 1 μl (50 μM) recombinant catalytic domain of the DUB enzyme USP2 (usp2 cc; E-504, Boston Biochem) if indicated.

### Immunoprecipitation

50 μl pre-washed and blocked (in 2.5% BSA) anti-FLAG M2 Agarose (A2220, Sigma) were rotated with 4.5 mg protein lysate overnight (single FLAG IP) or with 5.2 mg protein lysate for 2 h (double FLAG-GFP IP, FLAG IP of stably expressed CGRRF1-FLAG-HA and Hrd1-FLAG-HA) at 4°C. The beads were subsequently washed seven times with lysis buffer or TBST and rotated for 30 min at 4°C with 100 μl 3× FLAG peptide elution buffer (F4799, Sigma) according to manufacturer's conditions. The obtained eluate was mixed with SDS buffer (single FLAG IP) or used for subsequent GFP IP (double FLAG-GFP IP). For GFP IP, 40 μl pre-washed and blocked (in 2.5% BSA) GFP-TRAP beads (gta-20, Chromotek) were rotated with the eluate obtained from FLAG competition for 2 h (double IP) or with 3.8 mg protein lysate overnight (single GFP IP) at 4°C. The beads were washed seven times with TBST or lysis buffer and taken up in SDS sample buffer.

### Blue Sepharose

The original Blue Sepharose protocol (Ross *et al*, 2014) was modified according to Glaeser *et al*, 2016. Conditioned medium was recovered ~72 h after cell seeding and ~48 h after plasmid transfection to analyze the secretion of Wnt3A-KDEL or 48 h after reverse transfection and 24 h after plasmid transfection to analyze the effect of ERAD components. Conditioned medium was centrifuged at 2,000 *g* for 5 min and subsequently supplemented with Triton X-100 to a final concentration of 1% (v/v). In binding and wash buffer (150 mM KCl, 50 mM Tris–Cl, pH 7.5, 1% (v/v) Triton X-100) pre-washed Blue Sepharose 6 Fast Flow beads (17-0948-01, GE Healthcare) were added to the medium and rotated overnight at 4°C. The beads were washed twice with binding and wash buffer by centrifugation at 2,700 *g* for 5 min at 4°C, taken up in 50 μl reducing 1× SDS buffer and incubated at 96°C for 5 min. To assess the impact of VCP, CGRRF1, and UBE2J2 knockdown on Wnt secretion, it is important to not express the Wnt ligands longer than ~24 h since otherwise Evi is sufficiently stabilized through Wnt expression.

## Cell lysis and immunoblotting

Cells were lysed in 8 M urea for Evi stability assays or in eukaryotic lysis buffer (20 mM Tris–HCl, pH 7.4, 130 mM NaCl, 10% (w/v) glycerol, 2 mM EDTA, 1% (v/v) Triton X-100) supplemented with cOmplete™ Mini Protease Inhibitors (Roche) for immunoprecipitation and TUBE assays. For TUBE experiments, the lysis buffer was additionally supplemented with inhibitors of deubiquitinating enzymes: 5 mM N-ethylmaleimide, 100 μM PR-619, and 2 mM 1,10-phenanthroline. For FLAG IP of stably expressed CGRRF1-FLAG-HA and Hrd1-FLAG-HA, cells were lysed in 50 mM Tris pH 7.4, 150 mM NaCl, 5 mM EDTA, 1% (w/v) LMNG (Anatrace), supplemented with protease inhibitors. PageRuler Plus Prestained (26619, Life Technologies) and Precision Plus Protein All Blue Prestained (1610373, Bio-Rad) were used as protein ladders. Membranes were probed with the following antibodies: LRP6 (rab, CST, 2560), β-catenin (mouse, BD BioScience, 610154), V5 (rab, Rockland, 600-401-378), Wnt3A (rab, Abcam, ab28472; was used to detect Wnt3 and Wnt3A), N-cadherin (mouse, BD BioScience, 610921), Wnt5A (goat, R&D Systems, AF645; was used to detect Wnt5A and Wnt5B), Wnt16 (rab, GeneTex, GTX128468), Wnt11 (rab, GeneTex, GTX105971), β-actin (HRP-coupled, Santa Cruz, sc-47778), HA (mouse, Roche, 12CA5), K48-linkage-specific poly-ubiquitin (rab, NEB, 8081S), VCP (mouse, Abcam, ab11433), FLAG (mouse, Origene, TA50011), SEL1L (mouse, Abcam, ab78298), HSC70 (mouse, Santa Cruz, sc-7298), and the monoclonal (clone YJ5) Evi antibody (mouse, Biolegend, 655902). Polyclonal rabbit anti-Evi antibodies were generated using the synthetic peptides CHVDGPTEIYKLTRKEAQ for Western blot and FTSPKTPEHE-CRYYN for IHC (Augustin *et al*, 2012), respectively. HRP-coupled goat anti-mouse (Jackson ImmunoResearch, #115-035-003), goat anti-rabbit (Jackson ImmunoResearch, #111-035-003), and donkey anti-goat (Santa Cruz, sc-2020) were used as secondary antibodies. Adobe Photoshop was used to generate cutouts from the full scan Western blots (Source data) and to adjust the brightness and contrast of entire Western blot images of some panels. If indicated, Western blots were quantified using ImageQuant LAS4000 and the corresponding software.

## *In situ* RNA and immunohistochemistry

*In situ* RNA hybridization was performed using the RNAscope 2.5 HD Reagent Kit-RED (322350, Advanced Cell Diagnostic) according to manufacturer's protocol. Briefly, tissue slides were baked for 1 h at 60°C, deparaffinized with xylene (15869204, MP Biomedicals) and ethanol, and incubated with hydrogen peroxide for 10 min at RT. After target retrieval for 15 min at 97–100°C, dried tissue slides were treated with protease plus for 30 min at 40°C and hybridized with the appropriate probes for 2 h at 40°C. After several amplification steps in accordance with manufacturer's conditions, the signal was detected using the supplied Fast RED dye, counterstained with 50% hematoxylin (sc-396328A, Santa Cruz) and 0.02% ammonium hydroxide (5460.1, Carl Roth), and mounted with EcoMount (320409, Advanced Cell Diagnostics). The following RNAscope probes from Advanced Cell Diagnostic were used: Hs-Wls (NM_001193334.1, region 355-132; Cat#415851), DapB (EF191515, region 414-862; Cat#310043), Hs-PolR2A (NM_000937.4, region 2514-343; Cat#310451).

Immunohistochemistry was performed as described previously (Augustin *et al*, 2012) with some modifications. In brief, tissue sections were deparaffinized in xylene, passed through serial dilutions of ethanol, and rehydrated in PBS. Antigen unmasking was carried out by microwaving for 10 min at 96°C with 10 mM citrate buffer (pH 6.0). To block endogenous peroxidase, tissue sections were incubated with 3% $H_2O_2$ for 20 min, followed by incubation with avidin (45 min) and biotin (15 min) blocking solution at RT in a humid chamber. After overnight incubation at 4°C with primary Evi antibody (1:200 in PBS containing 5% milk and 5% goat serum), tissue sections were washed with PBS and incubated for 5 min with PBS containing 0.05% Triton X-100. Then, slides were incubated with secondary biotinylated antibody (1:200 in PBS containing 5% milk and goat serum; E0432) for 3 h at RT, followed by 1-h incubation with AB complex (VEC-PK-6100, Biozol Diagnostica). Slides were incubated with diamino-benzidine (Biozol Diagnostica- SK-4100) until sufficient signal was detectable and counterstained with Mayer's hemalum (T865, Carl Roth). After dehydration with a dilution series of ethanol and xylene, slides were mounted with Histofluid (Engelbrecht, 19350).

**Expanded View** for this article is available online.

## Acknowledgements

We are grateful to M. Lemberg, B. Shilo, N. Volkmar, and members of the Boutros laboratory for helpful discussions. We thank B. Rauscher for re-analyzing TCGA datasets and statistical advice, as well as J. Bageritz, M. Lemberg, and F. Port for critical comments on the manuscript. We would like to thank C. Niehrs, K. Basler, S. Özbek, J. Behrens, D. Virshup, M. Waterman, X. He, N. Dantuma, F. Zhang, D. Trono, E. Campeau, S. Johnson, and R. Adam for sharing reagents. We are also grateful to the tissue bank of the National Center for Tumor Diseases (NCT Heidelberg, Germany) for providing samples as well as to S. Schecher and the Tissue Bank of the University Medical Center Mainz for scanning tissue slides. K.G. was supported in part by a fellowship of the Helmholtz-Israel Graduate School. E.F. is supported by a fellowship from the Medical Research Council. J.C.C. is supported by a grant from the Medical Research Council (MR/L001209/1) and by the Ludwig Institute for Cancer Research. Work in the laboratory of M.B. is supported in part by the DFG.

## Author contributions

KG designed, performed, and analyzed experiments and wrote the manuscript; MU, DK, and OV contributed to experiments; EF, FL, and JCC designed experiments to identify E3 ligases; MB designed experiments and wrote the manuscript; all authors contributed to edit the manuscript.

## Conflict of interest

The authors declare that they have no conflict of interest.

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
