## [Review Process File · The EMBO Journal]

ERAD-dependent control of the Wnt secretory factor Evi

Kathrin Glaeser, Manuela Urban, Emma Fenech, Oksana Voloshanenko, Dominique Kranz, Federica Lari, John C. Christianson and Michael Boutros

Review timeline:	Submission date:	10 May 2017
	Editorial Decision:	13 June 2017
	Revision received:	16 October 2017
	Editorial Decision:	07 November 2017
	Revision received:	01 December 2017
	Accepted:	02 January 2018

Editor:

Transaction Report:

1st Editorial Decision

13 June 2017

Thank you for submitting your manuscript for consideration by the EMBO Journal. We have now received three referee reports on your manuscript, which are included below for your information.

As you will see, all reviewers express interest in the proposed mechanism of Evi stability regulation, and they appreciate the high quality of presented data. Based on the positive recommendations of the reviewers, I would like to invite you to submit your revised manuscript while addressing the comments of all referees, and particularly focusing on the following aspects:

- Provide further support for Wnt-independent regulation of Evi expression (referee #1, point 1),
- Clarification of the effect of Wnt3/3A on Evi protein levels (referee #1, point 2; referee #3, points 2 and 3)
- Provide information on ERAD component expression in colon cancer samples (referee #1, point 1; referee #2, point 1)
- Expand the analysis of the enzymes involved in Evi ubiquitination (referee #2, point 7)

We generally allow three months as standard revision time. Please contact us in advance if you would need an additional extension. As a matter of policy, competing manuscripts published during this period will not negatively impact on our assessment of the conceptual advance presented by your study. However, upon publication of any related work please contact me as soon as possible in order to discuss how to proceed.

Please feel free to contact me if you have any further questions regarding the revision. Thank you for the opportunity to consider your work for publication. I look forward to your revision.

 Referee #1:

The secretion of Wnt proteins is dependent on the cargo-receptor Evi (also known as Wntless),

which transports Wnt through the secretory pathway for release at the cell surface. In this manuscript, the authors make the interesting observation that the protein level of Evi is dependent on the level of Wnt expression. This effect is independent of *evi* transcription, and through an elegant set of experiments, the authors show that Evi protein levels are regulated through the ERAD protein degradation pathway, which targets Evi for proteasomal degradation when it is not bound to lipidated Wnt. This pathway ensures that the level of Evi is fine-tuned to the amount of Wnt that needs to be secreted.

This is a very interesting study that provides important new insight into the mechanism of Wnt secretion. Although the work is of high quality, I have a number of comments that the author need to address.

Comments:

-One of the key experiments in this study is showing that the expression of *evi* is independent of Wnt signaling. It has previously been shown that mouse *evi* (also known as Gpr177) is a target of the canonical Wnt/beta-catenin pathway (Fu et al. PNAS 2009), so the finding that human *evi* is not regulated by Wnt signaling is unexpected. Therefore, the authors have to be extra rigorous in showing that human *evi* is not a Wnt target gene. The evidence that they show is based on activating the Wnt/beta-catenin pathway through recombinant Wnt3A, Dvl2 overexpression or GSK3beta inhibition. And although there clearly is no effect on *evi* expression, the effect on the Wnt target gene *Axin2* (which serves as a positive control) is also very minor in case of recombinant Wnt3A or Dvl2 overexpression (Fig. S2b). Further evidence can be provided by showing that inhibition of Wnt/beta-catenin signaling (through knock down of beta-catenin or overexpression of dominant negative TCF) has no effect on *evi* either. Moreover, the authors should compare their results in human cells to mouse cells to address these conflicting findings.

-The increase in Evi protein levels in colon cancer is accompanied by a strong decrease in Wnt3 expression (Fig. S1a), which is not in agreement with their model. It would be very informative (and strengthen their conclusions) if the authors show that the increase in Evi can be explained by changes in expression of the ERAD components (VCP, UBE2J2, CGRRF1).

-In Fig. 4d, knock down of VCP strongly increases Wnt3/3A levels. What is the explanation for this effect and can the authors rule out the possibility that this increase in Wnt3/3A is responsible for the observed increase in Evi protein level?

-What is the evidence that HCT116 and A375 express high levels of Wnt (see page 6)? A direct comparison to a low Wnt expressing cells is needed to make the point that Evi levels are more dependent on porcupine in these cell lines. The authors indicate that there is a clear effect on Evi level at 2-4 hours in Fig. 2e, but this is not clear without quantification.

-Given the confusing background staining in the Evi Western blots, it would be extremely helpful if the results displayed in all Evi Western blots are quantified.

Referee #2:

This is a nice manuscript in which the authors determine endogenous factors that control the regulation of EVI, the Wnt cargo receptor. They provide convincing data to show that Evi protein levels are specifically mediated by Wnt ligands and not by Wnt pathway activation. In the absence of Wnt ligands, Evi is ubiquitinated and degraded in a p97 and proteasome dependent pathway. They postulate on the role of Porcn in triaging Evi for forward trafficking through the secretory pathway or retrograde trafficking for ERAD-mediated degradation. This is a nice demonstration of an important homeostatic mechanism of regulation of Wnt signaling.

In general the data are convincing, and it is admirable that the majority of work on Evi looks at endogenous levels of the protein, and not overexpression which is often the case in these sorts of studies. My major concern is that the data in Figure 6 claiming to have identified the E2 and E3 responsible is neither complete nor especially convincing. I would suggest that either this data is really firmed up, or alternatively removed - it would not be appropriate to leave in its present

condition.

Fig 1a Can they explain the observation of increased Evi protein in the absence of increased mRNA? - Now that they show that EVI levels are regulated by components of the ERAD machinery, are abnormalities in individual components identified in tumours?

Fig 1c The authors claim that: expression of all tested Wnt ligands led to an increase in Evi protein levels, which was not observed upon expression of secreted luciferase (sLuc) or IGFBP5-V5.

First, the Evi staining in these gels is horrible as it runs just underneath a large background band - making it very difficult to identify a true increase in Evi detection - but this isn't the case with many of the other Evi blots - why is this? Second, it clearly is NOT the case for some of the Wnt ligands (eg 8/11 and possible some others but the blot are too messy to tell). These observations are simply ignored in the text and figure legend? The authors need to report what they observe - not what they want the reader to think they observe.

Fig 2 - There is a clear demonstration using a number of different orthogonal approaches that the Wnt-induced increase in Evi protein levels was palmitoylation dependent ie completely blocked upon Porcn inhibition, indicating that Porcn activity is required for Wnt-mediated Evi regulation. Fig2E - difficult but clear data.

Fig 3 Should be noted that a common misunderstanding is that MG132 is a proteasome inhibitor - it isn't - it's a cysteine protease inhibitor - which also has activity against the proteasome. There should be at least one experiment showing that a bona fide proteasome inhibitor eg bortezomib or others shows the same effect as MG132 in figure 3. Ubiquitination experiment is convincing. Use a Wnt-KDEL mutant to show that palmitoylated Wnt proteins stabilize Evi in the ER - actually I would have preferred to have seen a pulse-chase experiment.

Fig 4 - Convincing evidence for the role of p97 - using both depletion of p97 (which is usually fairly toxic) as well as pharmacological inhibitors. Nice demonstration that inhibitors of porcupine lead to Evi degradation in a p97-dependent pathway.

Fig 5 Nice data to show that Evi interacts with Porcpn and this interaction appears to be Wnt independent as long as Evi degradation is inhibited. Evi also interacts with catalytically inactive p97. They go on to provide some evidence for a triage decision of Evi degradation unless Wnt is present.

Fig 6 attempts to identify components of the ubiquitin machinery required for Evi degradation. The approach is limited as they use a candidate gene approach rather than an unbiased approach. They identify UBE2J2 as a potential E2 conjugating enzyme and CGRRF1 as a potential E3 ligase involved in this pathway. This is potentially interesting. However, I have some real concerns about these data, and for example, their statement on page 10 that 'These results indicate that Evi poly-ubiquitination is specifically mediated by the E2 conjugating enzyme UBE2J2.' is clearly hugely overstating their findings! They then choose a set of E3s - some of which are known to be involved in ERAD - but some ERAD E3 ligases are also missing from their collection. Their focus on CGRRF1 is a little misleading as their data suggests that multiple ligases may be involved ie CGRRF1, MARCH6 and TRC8? They specifically focus on the first- probably because it is novel, and may have the largest phenotype. As with the evidence for E2J2, they make overarching statements which are clearly not substantiated by their experimental data. This is unfortunate as it does somewhat spoil what is otherwise a very nice story. It's clear from their ubiquitin block that neither E2J2 nor CGRRF1 is the only (or indeed the main E2 or E3 respectively) enzyme responsible for Evi ubiquitination, as the loss of ubiquitin is only partial. This doesn't mean that these two enzymes are not involved - just that there are other ubiquitin conjugating enzymes and E3 ligases which must also be playing a role here. I would suggest this data is either dramatically improved and they identify the ligases responsible - or that this section is removed - it spoils what is otherwise a very nice story.

Discussion

Are altered levels of the ERAD machinery responsible for the increase in Evi seen in tumours?

Their discussion of the E3 ligases implicated in ERAD pathways is inaccurate and misleading and should be removed or improved upon.

Referee #3:

In this study, Glaeser et al. showed that the protein level of Evi, which act as a Wnt carrier in secretion, is reduced by ER-associated degradation (ERAD). Wnt overexpression restored this reduction depending on Porcupine, O-acyltransferase for the Wnt family proteins. In the absence of Wnt, Evi is poly-ubiquitinated and degraded by the VCP-mediated ERAD system. The authors also identified the E2 conjugating enzyme, UBE2J2, and E3 ligase, CGRRF1, for this degradation. Finally, they showed ERAD-dependent degradation actually controls Wnt secretion by knockdown of VCP, UBE2J2, or CGRRF1.

This study provides novel findings for understanding of the regulatory mechanism of Wnt secretion as well as of the physiological role of ERAD-mediated protein degradation. Especially, this study clearly showed that Evi is degraded by ERAD and the inhibitory effect of Wnt proteins on this degradation. The experiments shown in this study were well explicated and support the main conclusions. Therefore, I believe that this paper meets stringent criteria for publication in EMBO Journal.

On the other hand, I also feel this paper still remains several points that could be clarified and improved.

1. The mechanism by which Wnt blocks the ubiquitination of Evi: The authors insist that a triaging complex of Porcn and VCP determines whether Evi enters the secretory or the ERAD pathway. The major basis of this argument is the results shown in Figure 5, in which physical interactions between Porcn, Evi, and VCP were examined. Although the results shown in this figure well support a formation of ternary complex of these components, it still remains unclear how Evi is directed into the Wnt secretory pathway by Wnt/Evi complex. Since it is known that lipidated Wnt can bind to Evi, it should be important to examine whether Wnt-bound form of Evi could form a ternary complex with Porc and VCP or not. As it might be possible that Wnt-associated Evi is easily moved out from the ER, ER-retained form of Wnt, KDEL-Wnt3a, would be useful for this experiment.

2. Ubiquitination of Evi upon Wnt expression shown in Figure 5: Whereas ubiquitination of Evi in protein complexes was reduced by Wnt3a expression in figure 5a and b, it was not reduced in Figure 5c. The authors should carefully consider this difference. If this difference is reproducibly detected, the authors should consider their model by taking this difference into account.

3. Evi instability even in Porc knockout cells: Figure 2c shows that Evi was unstable in Porc knock-out cells in the presence or absence of Wnt3a. This data suggest that Evi may be degraded not through formation of the Porcm/Evi/VCP ternary complex. How do the authors explain the consistency of this data and their model, shown Figure 7d and Supplemental Figure 5d?

Minor comments

(1) Figure 1a: Signals of Evi proteins in Colorectal Cancer are unclear.

(2) Supplemental Figure 1c mRNA: Labels on the figure indicates that 3 different probes were used although figure legend shows that in situ hybridization with Evi probe was performed by using samples of 3 patients. Which is true?

Response to reviewers

Manuscript #EMBOJ-2017-97311

We thank the reviewers for their helpful comments. We have addressed the comments by additional experiments and rewriting of the manuscript. To address the reviewers' comments we have extended our experimental analysis with the following experiments:

- Figure 1C: We used another, commercially available mouse monoclonal Evi antibody to show Evi stabilization with less background bands upon expression of different Wnt ligands (in response to reviewer 2).
- Figure EV1D, D': To further rule out that Evi is a transcriptional Wnt target gene in HEK293T cells, we showed that β -catenin knockdown did not affect Wnt-stabilized Evi protein levels (in response to reviewer 1).
- Figure EV2A: We included mRNA levels of Axin2 in this experiment to enable the comparison of Evi expression with a known canonical Wnt target gene, confirming that also under these conditions, Evi is not regulated through canonical Wnt signaling on transcriptional level (in response to reviewer 1).
- Figure EV3B: This additional experiment shows that Bortezomib, another proteasome inhibitor, particularly stabilized Evi in the absence of expressed Wnt ligands, which confirms the effect of MG132 treatment (in response to reviewer 2).
- Figure EV3C: TUBE-precipitated poly-ubiquitinated proteins were treated with deubiquitinating (DUB) enzymes to confirm that the detected high molecular weight Evi signal was caused by direct poly-ubiquitination of Evi.
- Figure 6E: We showed an interaction between endogenous Evi and stable expressed FLAG-HA tagged CGRRF1 to further confirm CGRRF1 as an Evi-processing enzyme (in response to reviewer 2).
- Figure EV5D: Mutation of two cysteine residues within the RING domain of CGRRF1 acts as substrate trap for Evi stabilizing its interaction with CGRRF1. Additionally, these CGRRF1 mutants increased the abundance of Evi protein levels indicating that an active RING domain of CGRRF1 is required for efficient Evi degradation (in response to reviewer 2).
- Figure EV5E: This additional experiment shows a direct interaction between CGRRF1 and UBE2J2 indicating that both proteins could be part of the same ubiquitination machinery (in response to reviewer 2).
- Figure EV5G, H: We analyzed TCGA data, which shows reduced expression of CGRRF1 in colorectal and endometrial cancer when compared to the healthy tissue (panel G). Reduced CGRRF1 levels and increased expression of Wnt ligands could be one possible explanation of increased Evi protein levels in the absence of enhanced Evi transcription (panel H) in these cancer types (in response to reviewer 1-3).

Specific responses to reviewer's comments are below.

Response to Referee #1:

"The secretion of Wnt proteins is dependent on the cargo-receptor Evi (also known as Wntless), which transports Wnt through the secretory pathway for release at the cell surface. In this manuscript, the authors make the interesting observation that the protein level of Evi is dependent on the level of Wnt expression. This effect is independent of evi transcription, and through an elegant set of experiments, the authors show that Evi protein levels are regulated

through the ERAD protein degradation pathway, which targets Evi for proteasomal degradation when it is not bound to lipidated Wnt. This pathway ensures that the level of Evi is fine-tuned to the amount of Wnt that needs to be secreted.

This is a very interesting study that provides important new insight into the mechanism of Wnt secretion. Although the work is of high quality, I have a number of comments that the author need to address.”

We thank the reviewer for the positive and encouraging comments.

“Comments:

-One of the key experiments in this study is showing that the expression of evi is independent of Wnt signaling. It has previously been shown that mouse evi (also known as Gpr177) is a target of the canonical Wnt/beta-catenin pathway (Fu et al. PNAS 2009), so the finding that human evi is not regulated by Wnt signaling is unexpected. Therefore, the authors have to be extra rigorous in showing that human evi is not a Wnt target gene. The evidence that they show is based on activating the Wnt/beta-catenin pathway through recombinant Wnt3A, Dvl2 overexpression or GSK3beta inhibition. And although there clearly is no effect on evi expression, the effect on the Wnt target gene Axin2 (which serves as a positive control) is also very minor in case of recombinant Wnt3A or Dvl2 overexpression (Fig. S2b). Further evidence can be provided by showing that inhibition of Wnt/beta-catenin signaling (through knock down of beta-catenin or overexpression of dominant negative TCF) has no effect on evi either. Moreover, the authors should compare their results in human cells to mouse cells to address these conflicting findings.”

We provide multiple lines of evidence (Fig 1D and EV1C; Fig 2B and EV2A) that the protein but not the mRNA levels of Evi were increased in the presence of Wnt ligands indicating that the Wnts affect Evi abundance not *via* its transcription. Additionally, we now included the expression levels of Axin2 in Fig EV2A to compare the expression of Evi with a known canonical Wnt target gene. In contrast to Evi expression, transcriptional regulation of Axin2 was increased upon Wnt3 expression and sensitive to Porcn inhibition (Fig EV2A).

Importantly, Evi protein levels were also increased upon expression of non-canonical Wnts, which do not induce β -catenin-dependent Wnt target gene expression (Fig 1C). These results indicate that human Evi is not transcriptionally regulated by canonical Wnt pathway activation in human HEK293T cells.

To further support these findings, we thank the reviewer for the useful suggestion to additionally analyze the effect of β -catenin knockdown on Evi regulation. Overexpression of Wnt3A increased Evi protein levels and induced canonical Wnt reporter activity. While silencing of β -catenin blocked canonical Wnt signaling activity (new Fig EV1D’), Evi protein levels were not affected (new Fig EV1D) confirming that Evi is not a canonical Wnt target gene.

While the analysis in different systems might be indeed interesting, our manuscript addresses the acute regulation of Evi as the Wnt cargo receptor by ERAD in human cells. We provide several independent lines of evidence to support our conclusions. This does not rule out additional levels of regulation, e.g. by transcriptional control during cell differentiation. However, we would like to emphasize that we found no evidence in a clearly controlled system for a direct regulation of Evi through ‘downstream’ Wnt signaling.

We would also like to point out that the evidence in Fu et al. (Fu et al, 2009) is rather indirect by either using an artificial luciferase reporter construct fused to the promoter region of Evi or by inserting a β -geo reporter into the Evi locus instead of directly analyzing Evi mRNA levels. Apart from these reporter constructs, Evi was investigated on protein level in Fu et al.

“-The increase in Evi protein levels in colon cancer is accompanied by a strong decrease in Wnt3 expression (Fig. S1a), which is not in agreement with their model. It would be very informative (and strengthen their conclusions) if the authors show that the increase in Evi can be explained by changes in expression of the ERAD components (VCP, UBE2J2, CGRRF1).”

Indeed, Fig EV1A (before called Fig. S1a) shows an increase rather than a decrease in Wnt3 expression in colon adenocarcinoma when compared to healthy colon and rectum, which is in agreement with our model. Additionally, CGRRF1 expression is reduced in colon and endometrial cancer (TCGA, new Fig EV5G), which- together with higher Wnt expression- could explain increased Evi protein levels in these cancer types without an elevated expression of Evi (TCGA, new Fig EV5H).

“-In Fig. 4d, knock down of VCP strongly increases Wnt3/3A levels. What is the explanation for this effect and can the authors rule out the possibility that this increase in Wnt3/3A is responsible for the observed increase in Evi protein level?”

Increased Wnt3/3A levels upon VCP knockdown could indicate that overexpressed Wnt3/3A is partly targeted for ERAD as well. However, we can rule out that the increased Evi protein levels upon VCP knockdown are solely due to the higher Wnt3/3A levels since (1) Evi is also stabilized upon VCP knockdown without Wnt3/3A overexpression (Fig 4B) and (2) since the same increase in Wnt3/3A levels upon VCP knockdown did not further increase Evi protein levels in the absence of LGK974 treatment (Fig 4D, lanes 3, 5). Additionally, we showed in Fig 2 that non-palmitoylated Wnt proteins (which is equivalent to LGK974 treatment in Fig 4D, lanes 8, 10) are not able to increase Evi protein abundance. Therefore, we are confident that the effects on Evi regulation in Fig 4D are due to VCP knockdown.

“-What is the evidence that HCT116 and A375 express high levels of Wnt (see page 6)? A direct comparison to a low Wnt expressing cells is needed to make the point that Evi levels are more dependent on porcupine in these cell lines. The authors indicate that there is a clear effect on Evi level at 2-4 hours in Fig. 2e, but this is not clear without quantification.”

HCT116 cells were reported to depend on Wnt secretion (Voloshanenko et al, 2013) and A375 cells were shown to secrete endogenous Wnt5A and Wnt10B and additionally express Wnt16 (Yang et al, 2012). We included this information in the manuscript and changed “2-4 hours” to “time-dependent manner”.

We used HEK293T cells as low Wnt expressing cells, which particularly showed dependence on Porcn upon Wnt3A overexpression (Fig 2B, C). This cell system allowed us to show the dependence of Evi on Porcn with and without high Wnt expression within the same genetic background (same cell line).

“-Given the confusing background staining in the Evi Western blots, it would be extremely

helpful if the results displayed in all Evi Western blots are quantified.”

We thank the reviewer for the suggestion and repeated the Western blots, where it could have been difficult to differentiate between actual Evi bands and background bands. In these experiments (Fig 1C; EV5D; 6E), we used another, commercially available anti-Evi mouse antibody (Biolegend, clone YJ5), which we termed Evi (YJ5) to distinguish from the originally used anti-Evi polyclonal rabbit antibody. Since we used films for most of the displayed Western blots, quantification would not be appropriate. Importantly, in our opinion, the differences in Evi protein levels are very strong and therefore do not need an extra quantification.

Response to Referee #2:

“This is a nice manuscript in which the authors determine endogenous factors that control the regulation of EVI, the Wnt cargo receptor. They provide convincing data to show that Evi protein levels are specifically mediated by Wnt ligands and not by Wnt pathway activation. In the absence of Wnt ligands, Evi is ubiquitinated and degraded in a p97 and proteasome dependent pathway. They postulate on the role of Porcn in triaging Evi for forward trafficking through the secretory pathway or retrograde trafficking for ERAD-mediated degradation. This is a nice demonstration of an important homeostatic mechanism of regulation of Wnt signaling.”

We thank the reviewer for the supporting comments on the manuscript.

“In general the data are convincing, and it is admirable that the majority of work on Evi looks at endogenous levels of the protein, and not overexpression which is often the case in these sorts of studies. My major concern is that the data in Figure 6 claiming to have identified the E2 and E3 responsible is neither complete nor especially convincing. I would suggest that either this data is really firmed up, or alternatively removed - it would not be appropriate to leave in its present condition.”

We are very thankful that the reviewer appreciates our work on ERAD in an endogenous setting. Since we aimed to dissect the regulatory role of ERAD, it was particularly important for us to analyze its effect on endogenous rather than on overexpressed Evi. We extended the revised manuscript and included new data showing:

- (1) A specific interaction between CGRRF1 and endogenous Evi (new Fig 6E)**
- (2) The involvement of CGRRF1's active RING domain in Evi regulation and Evi-CGRRF1 interaction (new Fig EV5D)**
- (3) An interaction between UBE2J2 and CGRRF1 (new Fig EV5E).**

“Fig 1a Can they explain the observation of increased Evi protein in the absence of increased mRNA? - Now that they show that EVI levels are regulated by components of the ERAD machinery, are abnormalities in individual components identified in tumours?”

The expression of Wnt ligands (e.g. Wnt3; Fig EV1A) is increased in different cancer types such as colorectal cancer. Additionally, CGRRF1 expression is reduced in colon and endometrial cancer (TCGA, new Fig EV5G). Increased Wnt and/or reduced CGRRF1 expression could explain the observation of increased Evi protein levels in the absence of an elevated transcription of Evi (TCGA, new Fig EV5H).

“Fig 1c The authors claim that: expression of all tested Wnt ligands led to an increase in Evi protein levels, which was not observed upon expression of secreted luciferase (sLuc) or IGFBP5-V5.”

“First, the Evi staining in these gels is horrible as it runs just underneath a large background band - making it very difficult to identify a true increase in Evi detection - but this isn't the case with many of the other Evi blots - why is this? Second, it clearly is NOT the case for some of the Wnt ligands (eg 8/11 and possible some others but the blot are too messy to tell). These observations are simply ignored in the text and figure legend? The authors need to report what they observe - not what they want the reader to think they observe.”

We repeated the Western blot of Fig 1C and used another, commercially available anti-Evi mouse antibody (Biolegend, clone YJ5). The originally used anti-Evi polyclonal rabbit antibody produced different background bands, depending on the number of times used for immunoblotting. Importantly, our data show that canonical and non-canonical Wnts are able to increase Evi protein levels. Indeed, in contrast to other non-canonical Wnts, Wnt11 seems to have less impact on Evi regulation. We have now more extensively described this on page 5 of the manuscript and have re-phrased our wording accordingly.

“Fig 2 - There is a clear demonstration using a number of different orthogonal approaches that the Wnt-induced increase in Evi protein levels was palmitoylation dependent ie completely blocked upon Porcn inhibition, indicating that Porcn activity is required for Wnt-mediated Evi regulation. Fig2E - difficult but clear data.”

We are very grateful for these comments, which appreciate our different approaches to demonstrate the important role of Wnt-palmitoylation and Porcn function on the Wnt-dependent Evi regulation.

“Fig 3 Should be noted that a common misunderstanding is that MG132 is a proteasome inhibitor - it isn't - it's a cysteine protease inhibitor - which also has activity against the proteasome. There should be at least one experiment showing that a bona fide proteasome inhibitor eg bortezomib or others shows the same effect as MG132 in figure 3. Ubiquitination experiment is convincing. Use a Wnt-KDEL mutant to show that palmitoylated Wnt proteins stabilize Evi in the ER - actually I would have preferred to have seen a pulse-chase experiment.”

As suggested by the reviewer, we repeated the experiment using Bortezomib. Comparable to MG132 treatment, Bortezomib addition stabilized Evi in the absence of Wnt ligands (new Fig EV3B). Nevertheless, although MG132 is not as selective as next-generation proteasome inhibitors such as Bortezomib or Carfilzomib, it still remains a potent and relatively specific inhibitor of proteasome activity at the concentrations used in our assays. We also tried to perform a pulse-chase assay for endogenous Evi, which was however not feasible with the used anti-Evi mouse antibody (Biolegend, clone YJ5). Since it was particularly important for us to investigate the turnover rates of Evi on endogenous level, we included a cycloheximide chase assay instead (Fig 4E).

“Fig 4 - Convincing evidence for the role of p97 - using both depletion of p97 (which is usually fairly toxic) as well as pharmacological inhibitors. Nice demonstration that inhibitors of porcupine lead to Evi degradation in a p97-dependent pathway.”

“Fig 5 Nice data to show that Evi interacts with Porcpn and this interaction appears to be

Wnt independent as long as Evi degradation is inhibited. Evi also interacts with catalytically inactive p97. They go on to provide some evidence for a triage decision of Evi degradation unless Wnt is present."

Many thanks for this positive statement.

"Fig 6 attempts to identify components of the ubiquitin machinery required for Evi degradation. The approach is limited as they use a candidate gene approach rather than an unbiased approach. They identify UBE2J2 as a potential E2 conjugating enzyme and GCRRF1 as a potential E3 ligase involved in this pathway. This is potentially interesting. However, I have some real concerns about these data, and for example, their statement on page 10 that 'These results indicate that Evi poly-ubiquitination is specifically mediated by the E2 conjugating enzyme UBE2J2.' is clearly hugely overstating their findings!

We apologize for this overstatement and realize that without *in vitro* reconstitution of Evi degradation, we cannot unequivocally state that UBE2J2 is specifically or exclusively the E2 ubiquitin-conjugating enzyme that mediates Evi turnover. We have rephrased this statement: "These results support the participation of UBE2J2 in the ubiquitination and degradation of Evi."

Nevertheless, we provide evidence that UBE2J2 is involved- perhaps not exclusively- in the regulation of Evi. We would like to point out that only a few E2s have been directly implicated in ERAD, the most prominent being UBE2J1, UBE2G2 and UBE2J2. In our experiment (Fig 6B), the most prominent effect was observed upon knockdown of UBE2J2, whereas depletion of other more conventional ERAD E2s did not affect Evi protein levels. As shown in Fig 6B and D, UBE2J2 knockdown increased Evi protein abundance and reduced its poly-ubiquitination profile.

So far, UBE2J2 has only been linked to ubiquitination with the non-canonical E3 TMEM129 (van de Weijer *et al*, 2014; van den Boomen *et al*, 2014), viral mK3 (Wang *et al*, 2009) and RNF185 (El Khouri *et al*, 2013). Our data expand these findings demonstrating the involvement of UBE2J2 in Evi degradation.

They then choose a set of E3s - some of which are known to be involved in ERAD - but some ERAD E3 ligases are also missing from their collection. Their focus on CGRRF1 is a little misleading as their data suggests that multiple ligases may be involved ie CGRRF1, MARCH6 and TRC8? They specifically focus on the first- probably because it is novel, and may have the largest phenotype. As with the evidence for E2J2, they make overarching statements which are clearly not substantiated by their experimental data. This is unfortunate as it does somewhat spoil what is otherwise a very nice story.

As Evi is an integral membrane protein and degraded from the ER (Fig 3D), we would expect a similar residency of the ubiquitination machinery, as it has been the case for most if not all regulated ERAD substrates. We have screened some of the ~25 E3s that have been reported to reside in the ER membrane (Neutzner *et al*, 2011), of which several have been characterized and others not (Claessen *et al*, 2012). Although silencing the ER-resident E3 ligases TRC8 and MARCH6 could have minor effects on Evi levels, as pointed out by the reviewer, the effect was weaker compared to CGRRF1 knockdown. As TRC8 and MARCH6 have been implicated previously in regulation of proteins related to cholesterol homeostasis, one possible explanation could be that their disruption might have pleiotropic effects on Evi stability.

In unpublished data being prepared for another manuscript (Fenech et al., in preparation), we carried out quantitative mass-spec proteomics of 21 ER-resident ubiquitin ligases to identify interacting proteins. Within this mass-spec approach, Evi was only found as a high-confidence interaction partner of CGRRF1, but neither of MARCH6 nor of TRC8. Accordingly, endogenous Evi co-precipitated with stable expressed FLAG-HA-tagged CGRRF1 but not with the archetypal ER-resident E3 Hrd1 (new Fig 6E). Importantly, in the present study we also showed that silencing of CGRRF1 using pooled and single siRNAs significantly stabilised Evi and reduced Evi poly-ubiquitination (Fig 6C, D).

It should also be noted that UBE2J2 affects the abundance and poly-ubiquitination of Evi (Fig 6B, D) and that neither TRC8 nor MARCH6 have been reported to use UBE2J2 as their E2.

It's clear from their ubiquitin block that neither E2J2 nor CGRRF1 is the only (or indeed the main E2 or E3 respectively) enzyme responsible for Evi ubiquitination, as the loss of ubiquitin is only partial. This doesn't mean that these two enzymes are not involved - just that there are other ubiquitin conjugating enzymes and E3 ligases which must also be playing a role here. I would suggest this data is either dramatically improved and they identify the ligases responsible - or that this section is removed - it spoils what is otherwise a very nice story."

We agree with the reviewer that multiple E3s may participate in Evi degradation (either in parallel or sequentially) as this has been demonstrated previously for other membrane-tethered model substrates (Bernasconi et al, 2010). To clarify this statement, we now explicitly mention on page 11 of the manuscript that additional E2 and E3 enzymes might be involved in the ubiquitination and regulation of Evi. However, it would be beyond this manuscript to systematically identify all E3s that might impact Evi degradation as many combinations may exist given the number of E3s and without a clear idea of what their individual substrate specificity may be. Our data demonstrate that both CGRRF1 and UBE2J2 are important for Evi degradation since their knockdown had profound effects on Evi abundance using multiple individual and pooled siRNAs (Fig 6B, C; EV5B, C). Reduction of polyubiquitinated Evi also correlated with the absence of either CGRRF1 or UBE2J2.

Residual ubiquitination of Evi upon CGRRF1 and UBE2J2 knockdown may be explained either by residual enzymatic activity of CGRRF1 and UBE2J2 as we are presenting knockdowns rather than knockouts or by additional E2s and E3s that may participate in parallel or sequentially in Evi processing. As stated above, we have rephrased our manuscript indicating the participation of CGRRF1 and UBE2J2 and acknowledging possible contributions of other E3s and E2s.

To further support the involvement of CGRRF1 in Evi regulation, we have now included data showing that a dominant-negative RING mutant of CGRRF1 traps Evi in its interactions and is able to increase Evi protein levels indicating that the active site of CGRRF1 is involved in Evi degradation (new Fig EV5D).

"Discussion

Are altered levels of the ERAD machinery responsible for the increase in Evi seen in tumours?"

Indeed, CGRRF1 expression appears to be reduced in colon and endometrial cancer (TCGA, new Fig EV5G), which- together with higher Wnt expression- could explain elevated Evi protein levels in these cancer types without an increase in Evi

expression (TCGA, new Fig EV5H).

“Their discussion of the E3 ligases implicated in ERAD pathways is inaccurate and misleading and should be removed or improved upon.”

We addressed this concern by rephrasing our manuscript acknowledging that besides of CGRRF1 and UBE2J2 additional E3s and E2s may be involved in Evi regulation. To further support the involvement of CGRRF1 and UBE2J2 in Evi degradation, we have provided additional data on (1) the specific interaction between CGRRF1 and endogenous Evi (new Fig 6E), (2) demonstrating the effects of the active RING domain in CGRRF1 on Evi regulation and Evi-CGRRF1 interaction (new Fig EV5D) as well as (3) showing an interaction between UBE2J2 and CGRRF1 (new Fig EV5E).

Response to Referee #3:

“In this study, Glaeser et al. showed that the protein level of Evi, which act as a Wnt carrier in secretion, is reduced by ER-associated degradation (ERAD). Wnt overexpression restored this reduction depending on Porcupine, O-acyltransferase for the Wnt family proteins. In the absence of Wnt, Evi is poly-ubiquitinated and degraded by the VCP-mediated ERAD system. The authors also identified the E2 conjugating enzyme, UBE2J2, and E3 liagase, CGRRF1, for this degradation. Finally, they showed ERAD-dependent degradation actually controls Wnt secretion by knockdown of VCP, UBE2J2, or CGRRF1.

This study provides novel findings for understanding of the regulatory mechanism of Wnt secretion as well as of the physiological role of ERAD-mediated protein degradation. Especially, this study clearly showed that Evi is degraded by ERAD and the inhibitory effect of Wnt proteins on this degradation. The experiments shown in this study were well explicated and support the main conclusions. Therefore, I believe that this paper meets stringent criteria for publication in EMBO Journal.

On the other hand, I also feel this paper still remains several points that could be clarified and improved.”

We are very thankful for these nice comments on our manuscript and hope to adequately address the remaining minor concerns of the reviewer.

“1. The mechanism by which Wnt blocks the ubiquitination of Evi: The authors insist that a triaging complex of Porcn and VCP determines whether Evi enters the secretory or the ERAD pathway. The major basis of this argument is the results shown in Figure5, in which physical interactions between Porcn, Evi, and VCP were examined. Although the results shown in this figure well support a formation of ternary complex of these components, it still remains unclear how Evi is directed into the Wnt secretory pathway by Wnt/Evi complex. Since it is known that lipidated Wnt can bind to Evi, it should be important to examine whether Wnt-bound form of Evi could form a ternary complex with Porc and VCP or not. As it might be possible that Wnt-associated Evi is easily moved out from the ER, ER-retained form of Wnt, KDEL-Wnt3a, would be useful for this experiment.”

Thanks for the interesting comment. Indeed, we could not only detect endogenous Evi, but also Wnt3A after performing the sequential Porcn-VCP IP, which indicates that both proteins are part of the Porcn-VCP complex. However, we cannot directly conclude whether the detected Wnt3A was bound to Evi.

The investigation if the Wnt-bound Evi is part of the ternary complex would require sequential immunoprecipitation of endogenous Evi and Wnt3A and subsequent blotting for VCP and Porcn. However, a sequential Wnt3A-Evi IP is currently not possible since we do not have the corresponding peptide of Evi or Wnt3A for competition. Nevertheless, it would be an interesting experiment for future studies but it is beyond the scope of the current manuscript.

However, as mentioned in the discussion of the manuscript (page 14), "Porcn may assist with the assembly of protein complexes in the ER such as an Evi-Wnt complex ... to support either the secretion of the fully assembled complex or the degradation of the remaining subunits". Such model would presume that the Wnt-bound Evi might- at least shortly- interact with the Porcn-VCP containing complex before being channeled into the secretory route *via* Porcn.

"2. Ubiquitination of Evi upon Wnt expression shown in Figure 5: Whereas ubiquitination of Evi in protein complexes was reduced by Wnt3a expression in figure 5a and b, it was not reduced in Figure 5c. The authors should carefully consider this difference. If this difference is reproducibly detected, the authors should consider their model by taking this difference into account."

It is an interesting observation. However, we are not sure if we can draw quantitative conclusions on the degree of Evi ubiquitination after performing a double IP experiment.

"3. Evi instability even in Porc knockout cells: Figure 2c shows that Evi was unstable in Porc knock-out cells in the presence or absence of Wnt3a. This data suggest that Evi may be degraded not through formation of the Porcm/Evi/VCP ternary complex. How do the authors explain the consistency of this data and their model, shown Figure 7d and Supplemental Figure 5d?"

To clarify this point, we rephrased our description of the model in our manuscript. However, we would like to emphasize that we do not claim that Porcn is needed for the degradation of Evi but rather for preventing its degradation through ERAD. Consequently, knockout of Porcn, treatment with the Porcn-inhibitor LGK974 or using a palmitoylated mutant form of Wnt3A resulted in the degradation of Evi preventing its Wnt3A-mediated stabilization (Fig 2).

The model of a triaging complex containing Evi, Porcn and VCP would allow to directly route Evi either into the secretory pathway (through Porcn) or into the ERAD pathway (through VCP), depending on its need for Wnt secretion. However, we do not think, that VCP and Porcn necessarily require each other to execute their function, which is inducing degradation of Evi (in case of VCP) or guiding Evi into the secretory pathway in the presence of Wnts (in case of Porcn). We tried to make this distinction clear in our abstract figure by using transparent colors for VCP in case of Porcn-dependent guidance of Evi into the secretory route and likewise for Porcn in case of VCP-dependent degradation of Evi through ERAD. Additionally, we described it in more detail in the legend of Fig 7D as well as in the discussion of the manuscript (page 13).

The idea that Porcn does not necessarily need VCP for its effect on Evi regulation is supported by Fig 5A showing a clear interaction between Evi and Porcn in the absence of VCP (VCP knockdown). Additionally, Wnt3A-and Wnt5A secretion was even increased upon VCP-knockdown (Fig 7A, B) indicating that Porcn function and guiding of Evi into the secretory route did not require VCP. Similarly, VCP does not

require Porcn activity to induce the degradation of Evi, which is supported by Fig 4D, showing that Evi is particularly degraded in a VCP-dependent manner when Porcn activity is blocked.

“Minor comments

(1) Figure 1a: Signals of Evi proteins in Colorectal Cancer are unclear.”

In contrast to the normal colon, colorectal cancer samples have a strong positive brown staining for Evi protein. We clarified this point in the Figure legends.

“(2) Supplemental Figure 1c mRNA: Labels on the figure indicates that 3 different probes were used although figure legend shows that in situ hybridization with Evi probe was performed by using samples of 3 patients. Which is true?”

We rephrased our figure legends to clarify this point. We analyzed subsequent slides of healthy colon and colon cancer tissue derived from the same patients. Of these subsequent slides, one slide of the healthy and the cancer tissue was stained for Evi protein, one for Evi mRNA (Fig 1A), one for DapB mRNA and one for PolR2A mRNA (Suppl. Fig 1A). PolR2A served as positive and DapB as negative control. The tissues of five individual patients were analyzed. Fig 1A and Suppl. Fig 1A is representative for three (out of five) patients, with some variation or lower increase observed in the other IHC stainings.

References

- Bernasconi R, Galli C, Calanca V, Nakajima T & Molinari M (2010) Stringent requirement for HRD1, SEL1L, and OS-9/XTP3-B for disposal of ERAD-LS substrates. *J. Cell Biol.* **188**: 223–235
- van den Boomen DJH, Timms RT, Grice GL, Stagg HR, Skødt K, Dougan G, Nathan JA & Lehner PJ (2014) TMEM129 is a Derlin-1 associated ERAD E3 ligase essential for virus-induced degradation of MHC-I. *Proc. Natl. Acad. Sci. U. S. A.* **111**: 11425–30
- Claessen JHL, Kundrat L & Ploegh HL (2012) Protein quality control in the ER: balancing the ubiquitin checkbook. *Trends Cell Biol.* **22**: 22–32
- Fu J, Jiang M, Mirando AJ, Yu H-MI & Hsu W (2009) Reciprocal regulation of Wnt and Gpr177/mouse Wntless is required for embryonic axis formation. *Proc. Natl. Acad. Sci. U. S. A.* **106**: 18598–603
- El Khouri E, Le Pavec G, Toledano MB & Delaunay-Moisan A (2013) RNF185 is a novel E3 ligase of endoplasmic reticulum-associated degradation (ERAD) that targets cystic fibrosis transmembrane conductance regulator (CFTR). *J. Biol. Chem.* **288**: 31177–31191
- Neutzner A, Neutzner M, Benischke A, Ryu S, Frank S, Youle RJ & Karbowski M (2011) A systematic search for endoplasmic reticulum (ER) membrane-associated RING finger proteins identifies Nixin/ZNRF4 as a regulator of calnexin stability and ER homeostasis. *J. Biol. Chem.* **286**: 8633–43
- Voloshanenko O, Erdmann G, Dubash TD, Augustin I, Metzsig M, Moffa G, Hundsrucker C, Kerr G, Sandmann T, Anchang B, Demir K, Boehm C, Leible S, Ball CR, Glimm H, Spang R & Boutros M (2013) Wnt secretion is required to maintain high levels of Wnt activity in colon cancer cells. *Nat. Commun.* **4**: 2610
- Wang X, Herr RA, Rabelink M, Hoeben RC, Wiertz EJHJ & Hansen TH (2009) Ube2j2 ubiquitinates hydroxylated amino acids on ER-associated degradation substrates. *J. Cell*

Biol. **187**: 655–668

van de Weijer ML, Bassik MC, Luteijn RD, Voorburg CM, Lohuis MAM, Kremmer E, Hoeben RC, LeProust EM, Chen S, Hoelen H, Rensing ME, Patena W, Weissman JS, McManus MT, Wiertz EJHJ & Lebbink RJ (2014) A high-coverage shRNA screen identifies TMEM129 as an E3 ligase involved in ER-associated protein degradation. *Nat. Commun.* **5**: 3832

Yang PT, Anastas JN, Toroni RA, Shinohara MM, Goodson JM, Bosserhoff AK, Chien AJ & Moon RT (2012) WLS inhibits melanoma cell proliferation through the β -catenin signalling pathway and induces spontaneous metastasis. *EMBO Mol. Med.* **4**: 1294–1307

2nd Editorial Decision

07 November 2017

Thank you for submitting a revised version of your manuscript. The manuscript has now been seen by the three original referees, who find that all their main concerns have now been addressed. There are just a couple of minor editorial issues to be dealt with before formal acceptance here. Congratulations on a nice study!

Referee #1:

The authors have addressed all of my concerns and I now fully support publication of this interesting and important manuscript.

Referee #2:

The authors have addressed my previous concerns. In particular they present more convincing evidence for the role of UBE2J2 and the E3 ligase CGRRF1 in the regulation of Evi. They have tempered their results and discussion accordingly. My other comments on the manuscript have been appropriately addressed.

Referee #3:

The revised manuscript was improved and I consider that there is no problem for publication of this manuscript in the EMBO journal.

2nd Revision - authors' response

01 December 2017

Authors incorporated the requested editorial changes.

Corresponding Author Name: Michael Boutros

Manuscript Number: EMBOJ 97311